# UserRL: Training Interactive User-Centric Agent via Reinforcement Learning

## Abstract

Reinforcement learning (RL) has shown promise in training agentic models that move beyond static benchmarks to engage in dynamic, multi-turn interactions. Yet, the ultimate value of such agents lies in their ability to assist users, a setting where diversity and dynamics of user interaction pose challenges. In this work, we propose **UserRL**, a unified framework for training and evaluating user-centric abilities through standardized gym environments paired with simulated users. We systematically vary turn-level reward assignment and trajectory-level score calculation to analyze how different formulations affect learning under the GRPO algorithm. Our experiments across Qwen3 models reveal that: (i) SFT cold start is critical for unlocking initial interaction ability and sustained RL improvements; (ii) deliberate trajectory scoring yields more efficient and effective multi-turn interactions; and (iii) while stronger simulated users (e.g., GPT-4o) facilitates training, open-source simulators (e.g., Qwen3-32B) remain a cost-effective and transferable option. Together, these results highlight that careful design of reward shaping and user simulation choice is as crucial as model scale, and we establish UserRL as a practical pathway for developing robust user-centric agentic models.

## 1 Introduction

Reinforcement learning (RL) has emerged as a powerful approach for training *agentic* large language models (LLMs), offering greater generalizability than supervised fine-tuning (Chu et al., 2025). This advantage is crucial for agents, which must operate in varied and unpredictable environments, solving tasks that often require extended reasoning and adaptation (Xi et al., 2023). A key enabler in this direction is multi-turn rollout, where the agent engages in multiple steps of interaction with its environment, producing rich trajectories that RL algorithms can exploit. For tool-using agents, multi-turn rollout is not optional but necessary: many real tasks demand sequences of tool calls, intermediate reasoning steps, and iterative refinements, in contrast to problems that can be solved in a single step (Deng et al., 2024b; Qian et al., 2025a).

Recent progress combining RL with multi-turn rollouts has produced increasingly capable agents (Wang et al., 2025; Zeng et al., 2025). These systems can coordinate multiple tool uses, navigate diverse domains such as web, code and games, and execute multi-stage reasoning pipelines (Deng et al., 2024a; Qian et al., 2024a; Zhu et al., 2025). Examples include iterative searches for open-domain QA (Jin et al., 2025), stepwise debugging for programming challenges (Golubev et al., 2025), and staged decision-making for complex browsing tasks (Lù et al., 2024). Such abilities mark a significant step toward agents that function as general problem-solvers.

However, an agent's ultimate value is not determined by its performance in abstract benchmarks or closed environments, but by its effectiveness in assisting **users**. Whether the context is scientific research, professional analysis, or everyday information gathering, the agent's role is collaborative (Baek et al., 2024; Sun et al., 2025; Zhang et al., 2025). This reframing shifts the perspective: the user is not merely a goal-setter or evaluator, but an integral and dynamic part of the agent's operating environment. The most capable agent is one that can understand, adapt to, and actively support the user throughout the task (Qian et al., 2024b; Lu et al., 2025).

Placing robust user assistance at the center exposes two critical interaction traits for training:

- **Diversity**: User behavior is heterogeneous and shaped by individual preferences, goals, and communication styles. This diversity demands that agents master a broad range of interaction skills.

- **Dynamics**: User interaction unfolds over multiple turns and can shift in intent or constraints as the conversation progresses, making pre-collected datasets unable to fully capture evolving patterns.

These traits, while essential to real-world assistance, create challenges for model training: there is no standardized framework to represent diverse user abilities, and it is difficult to simulate realistic, dynamic interactions within existing training pipelines. This leads to our core research question: *How can we design and train agentic models that effectively acquire user-centric abilities, while accounting for the inherent diversity and dynamics of user interactions?*

To address *diversity*, we design a **unified suite of user-centric gym environments**, each targeting distinct interaction skills and supporting both benchmarking and RL training. A standardized interface and customizable reward specification allow the environments to be adapted or extended for new scenarios. To address *dynamics*, we integrate **multi-turn RL rollouts** with **LLM-based user simulation**, enabling the agent to engage with adaptive, context-aware simulated users during training. These simulations provide realistic, evolving feedback, better approximating the complexity of live user interactions.

Empirically, our study leverages this setup to examine how to best supervise agents under the GRPO algorithm (Guo et al., 2025). The dense incremental reward signals provided by our gym environments make it possible to investigate two key aspects of reward shaping during multi-turn rollouts: (1) strategies for aggregating trajectory-level scores and (2) methods for assigning turn-level rewards. Across 4B and 8B Qwen3 models, we find that trajectory-level scoring proves more decisive than fine-grained turn differentiation. SFT cold start is also critical, preventing RL from early plateaus and enabling over 100% gains in some gyms. Models trained with weaker simulated users (Qwen3-32B) transfer well to stronger evaluators (GPT-4o), while stronger users still accelerate learning and boost performance. Beyond simulation, our models show greater interaction efficiency and leverage multi-turn interactions more effectively. In real-user evaluations, they can even surpass simulated user performance, thanks to the cooperative guidance human naturally offers. These results underscore that reward shaping and user simulation are as crucial as scale, establishing **UserRL** as a robust framework for user-centric agent training. We summarize our contributions as follows:

- We introduce a unified set of user-centric gyms with simulated users for dynamic, multi-turn engagement, enabling systematic benchmarking and training of diverse interaction abilities.
- We standardize the gym's interaction through a tool interface to support customization and future extension, making it convenient and scalable for RL pipelines.
- We comprehensively analyze user-centric RL through turn and trajectory-level reward shaping, highlighting design choices that enhance interaction efficiency and effectiveness.

We view our work as a step toward agents that are not merely task-solvers, but adaptive partners capable of actively understanding, reasoning, and collaborating with users in complex settings.

## 2 GYM CONSTRUCTION

Gymnasium environments have long been central to RL training (Towers et al., 2024). Their enduring value lies in providing a clean and reproducible interface through which agents can interact with environments. Building on this foundation, we construct eight novel gym environments, unified under a standardized interface, each with unique emphasis on user interaction. This design ensures that the environments not only support agent training and benchmarking, but also capture the nuances of real-world interaction. In the followings, we detail the principles underlying our construction.

**General Gym Components.** Each gym environment is built around two core components: the *task* and the *user*. From the **task perspective**, the environment can be seen as a finite automaton. After initialization through the `reset()` function, the environment transitions step-by-step based on agent actions, where each `step()` updates the internal state according to deterministic, rule-based transition functions. Along with the new state, the environment emits rewards that reflect whether progress toward task completion has been made. This structure ensures that the evaluation remains rigorous, transparent, and reproducible.

From the **user perspective**, certain agent actions are interpreted as input to the simulated user. The environment then returns a response generated by a LLM, which acts as the user simulator. Depending on the specific task, this feedback may take the form of a conversational utterance, a preference judgment, or an answer to a posed query. By employing LLMs, the user responses remain dynamic

Figure 1: The UserRL framework: Applying the standardized interact tool as interface, the policy model interacts with multiple Gym environments in the multi-turn rollout, generating groups of trajectories with turn-level rewards. A custom reward calculator remaps each trajectory into (i) a single trajectory-level score for advantage estimation and (ii) turn-level rewards, which are scaled and integrated to produce the final token-level advantages for policy updates.

and contextually adaptive, while the underlying task completion remains strictly rule-based. This dual design introduces both the rigor of formal evaluation and the richness of natural user interaction.

**Standardized Tool Interface.** A key innovation in our construction is the *standardized tool interface*, which governs how agents interact with the environment. As shown in Figure 1, this interface reduces all interactions to three core operation types: `Action`, `Search`, and `Answer`.

- **Action:** Direct communication with the simulated user. The agent provides utterances as input, and the gym returns user responses.
- **Search:** Retrieval of external knowledge. The agent issues a search query, and the gym interacts with a backend to provide retrieved content.
- **Answer:** Submission of a candidate solution. The agent provides a answer, and the gym verifies correctness if the task is goal-oriented.

This interface is deliberately minimal yet expressive: it captures the essential modes of agent behavior, keeps implementation simple, and allows straightforward extensibility. Importantly, the set of available options may vary by environment. For instance, TurtleGym permits all three operation types, while PersuadeGym restricts agents to `Action`, as persuasion tasks lack verifiable answers and completion is instead defined by user attitude change.

**Specific Gym Designs.** We design eight distinct gym environments, each targeting a different aspect of agent ability. The details are summarized in Table 1. Some gyms are newly curated, while others adapt existing benchmarks under our unified interface, demonstrating both the flexibility and extensibility of our framework. These environments collectively test a spectrum of user-centric skills: from intent understanding and persuasive communication, to personalized planning and tool use. They also probe different reasoning capacities, including creative, strategic, and mathematical reasoning. Taken together, our gyms provide a principled yet dynamic platform for advancing agent training and evaluation, ensuring that agents are not only competent in abstract tasks but also genuinely helpful in supporting users. For additional construction details, user simulation prompts and task design logic, please refer to Appendix C.

## 3 USERRL FRAMEWORK

Along with the gyms we release, we further present a reinforcement learning framework for training agents in user-centric environments. As illustrated in Figure 1 (right), UserRL extends GRPO by introducing a *flexible mechanism* for distributing rewards across turns and defining trajectory-level scores. Instead of fixing a single scheme, we provide a general interface where different strategies can be trialed, thereby opening room for empirical comparison and task-specific adaptation.

| Gym Name | Description | Data Source | Capability | Interface |
|---|---|---|---|---|
| IntentGym | Reveal user's real intent from agentic tasks | IN3 (Qian et al., 2024b) | Intent understanding, ambiguity resolution | Action: ask clarifying question |
| TurtleGym | Play turtle soup game with user to reveal hidden twists | Newly curated | Creative reasoning, contextual adaptation | Action: inquire story details; Answer: unveil the hidden story |
| PersuadeGym | Persuade user with opposing claims and arguments | Persuasion (Durmus et al., 2024) | Strategic reasoning, persuasive communication | Action: provide persuasive arguments |
| TelepathyGym | Guess out what entity user is thinking | Newly curated | Strategic reasoning, hypothesis testing | Action: interact for clues; Answer: guess out the entity |
| FunctionGym | Reveal user's hidden mapping rule for a set of numbers | Newly curated | Math reasoning, pattern generalization | Action: get number's mapping results; Search: retrieve test case; Answer: give test case result |
| TravelGym | Help user make personalized travel booking | UserBench (Qian et al., 2025b) | Preference elicitation, personalized planning | Action: interact for preferences; Search: retrieve travel options; Answer: provide travel recommendation |
| TauGym | Fulfill user requirements through tools and conversation | Tau-Bench (Yao et al., 2024) | Tool use, task-oriented interaction | Action: interact for user details; Search: retrieve available tools; Answer: perform tool call for task solving |
| SearchGym | Search and answer general user questions | Bamboogle (Press et al., 2022) | General helpfulness and reasoning | Search: retrieve online information; Answer: respond to user query |

Table 1: Eight gym environment details including data source, tested capability, and interface design.

| Gym | Travel | Turtle | Function | Tau | Persuade | Intention | Telepathy | Search | Total |
|---|---|---|---|---|---|---|---|---|---|
| Train Num. | 925 | 423 | 460 | 500 | 378 | 380 | 360 | 0 | **3426** |
| Test Num. | 471 | 48 | 78 | 165 | 42 | 40 | 41 | 125 | **1010** |
| Metric | Choice Correctness (Follow UserBench) | Sum Turn Reward | Test Case Correctness | Task Completion (Follow Tau-Bench) | Sum Turn Reward | Sum Turn Reward | Final Guess Correctness | Final Answer Exact Match | - |

Table 2: Statistics of eight gym environments, including data point numbers and metrics for evaluation.

**Multi-Turn Rollouts and Notation.** Let $\pi_\theta$ denote the policy and $T$ the number of interaction turns. A *multi-turn rollout trajectory* is

$$\tau = \{(s_1, a_1, r_1), (s_2, a_2, r_2), \ldots, (s_T, a_T, r_T)\},$$

where $a_t \sim \pi_\theta(\cdot \mid s_t)$, the environment transitions to $s_{t+1}$, and emits a *turn reward* $r_t$. Each turn $t$ corresponds to a generated sequence of tokens $x_t = (x_{t,1}, \ldots, x_{t,L_t})$.

*Motivation.* In multi-turn interaction, feedback is naturally incremental: some turns are exploratory, others solve subgoals, and later turns may finalize the outcome. Our gyms provide turn-level rewards, making it possible to use these dense signals for assigning credit across turns. This motivates a framework that can flexibly redistribute supervision and evaluate different hypotheses about what matters most in user interaction.

**Turn-Level Reward Shaping.** The gym produces raw turn rewards $\{r_t\}_{t=1}^T$. Before broadcasting them to tokens, we transform them into turn-level signals $\{\tilde{r}_t\}_{t=1}^T$:

$$R(x_{t,k}) = \tilde{r}_t, \quad \forall k \in \{1, \ldots, L_t\}.$$

This ensures that tokens within a turn share the same signal, while different turns can be treated differently depending on the chosen scheme.

We experiment with the following reward shaping methods:

- **Naive:** $\tilde{r}_t = r_t$, rewards remain unchanged, but in practice this often makes them too sparse, which quickly leads to training collapse.

- **Equalized:** $\tilde{r}_t = c$, a constant reward is assigned to every turn, effectively treating them all equally. This mirrors the approach used in the original GRPO, where each turn's reward is uniform.

- **Reward-to-Go (R2G):** Each turn accumulates discounted future rewards:

$$\tilde{r}_t = \sum_{j=t}^{T} \gamma^{j-t} r_j, \quad \gamma \in [0, 1].$$

- **Exponential Mapping (EM):** A nonlinear rescaling of $r_t \in [0, 1]$ into $[0.5, 1]$:

$$\tilde{r}_t = \phi_k(r_t) = 0.5 + 0.5 \cdot \frac{1 - \exp(-kr_t)}{1 - \exp(-k)}, \quad k > 0.$$

*Insight.* Each scheme reflects a different inductive bias: equalization emphasizes structural importance of all turns; reward-to-go propagates credit temporally, rewarding early enabling moves; exponential mapping ensures small positive progress is not lost, while still differentiating high rewards. The key point is that our framework makes it straightforward to trial such alternatives without altering the base optimization framework.

**Trajectory-Level Scoring.** GRPO requires a single scalar trajectory score for group-wise normalization. Since our environments produce incremental feedback, we define two strategies:

- **Sum:** As turn rewards reflect incremental gains, summing naturally recovers total progress.
- **Reward-to-Go (R2G):** This variant encourages efficient strategies that achieve progress earlier within fewer turns.

$$R_{\text{traj}}^{\text{sum}}(\tau) = \sum_{t=1}^{T} r_t, \qquad R_{\text{traj}}^{\text{r2g}}(\tau) = \sum_{j=1}^{T} \gamma^{j-1} r_j.$$

*Insight.* By exposing multiple scoring rules, our design allows principled exploration of how to best aggregate incremental user feedback. One may view the sum as reflecting raw task completion, while reward-to-go adds temporal preference.

**Grouped Advantage Estimation and Objective.** For each query $Q$, a rollout group $G_Q = \{\tau^{(i)}\}_{i=1}^{n}$ is collected. Using a chosen trajectory scorer $R_{\text{traj}}$, we compute:

$$\mu_Q = \tfrac{1}{n} \sum_{i=1}^{n} R_{\text{traj}}(\tau^{(i)}), \quad \sigma_Q = \sqrt{\tfrac{1}{n} \sum_{i=1}^{n} \left(R_{\text{traj}}(\tau^{(i)}) - \mu_Q\right)^2}.$$

Each token $x_{t,k}$ in trajectory $i$ is then assigned a normalized advantage:

$$A(x_{t,k}|Q) = \frac{\tilde{r}_t^{(i)} - \mu_Q}{\sigma_Q + \eta}, \quad \eta > 0.$$

*Objective.* This design keeps GRPO's normalization across comparable rollouts, but leaves the definition of per-turn and trajectory-level rewards flexible. This modularity allows us to test different biases, while keeping the optimization pipeline consistent. Finally, the policy is trained with the clipped PPO objective, adapted to our advantage formulation. Note that we also omit the KL loss in original objective to encourage exploration and alignment to our new interface:

$$J_{\text{UserRL}}(\theta) = \mathbb{E}_{Q \sim \mathcal{D}} \mathbb{E}_{\tau \sim \pi_{\text{old}}} \left[ \frac{1}{\sum_{t=1}^{T} L_t} \sum_{t=1}^{T} \sum_{k=1}^{L_t} \min\left( \rho_{t,k} A(x_{t,k}|Q), \, \text{clip}(\rho_{t,k}, 1 - \epsilon, 1 + \epsilon) A(x_{t,k}|Q) \right) \right],$$

where $\rho_{t,k} = \pi_\theta(x_{t,k} \mid \text{context}_{t,k})/\pi_{\text{old}}(x_{t,k} \mid \text{context}_{t,k})$.

*Summary.* UserRL generalizes GRPO to multi-turn interactive settings by decoupling *turn-level reward shaping* from *trajectory-level scoring*. Importantly, this design reflects *user-centric*: since the gyms provide turn-wise feedback grounded in user interaction, the framework is explicitly tailored to reflect user experience over time. By treating interaction as incremental progress rather than a single end-state, UserRL supports agents in learning behaviors that align with user needs, while giving researchers the flexibility to decide how such feedback should be weighted and aggregated.

## 4 EXPERIMENTS

Building on the framework and the gym environments we constructed, we now investigate what constitutes an effective RL setting for user-centric agentic tasks. Our analysis focuses specifically on *trajectory-level scoring* and *turn-level reward shaping*, two dimensions that are uniquely enabled by multi-turn, dense feedback from user-centric environments. These settings allow us to probe how different reward structures influence policy learning in interactive contexts.

| Model | TravelGym | TurtleGym | FunctionGym | TauGym | PersuadeGym | *IntentionGym* | *TelepathyGym* | *SearchGym* | Avg. |
|---|---|---|---|---|---|---|---|---|---|
| *Open-Source (Trained Model)* | | | | | | | | | |
| Qwen3-8B (Equalized/R2G) | **0.5730** | 0.1854 | **0.4231** | 0.1818 | 0.5317 | 1.8175 | 0.5610 | 0.8880 | **0.5652** |
| Qwen3-8B (EM/R2G) | 0.5025 | 0.1917 | 0.4103 | 0.2000 | **0.5397** | **1.9025** | 0.5366 | 0.8640 | 0.5343 |
| Qwen3-8B (R2G/R2G) | 0.5724 | 0.1615 | **0.4231** | 0.1394 | 0.5238 | 1.8525 | 0.5854 | 0.8480 | 0.5539 |
| Qwen3-8B (Equalized/Sum) | 0.5054 | 0.1323 | 0.2692 | **0.2121** | 0.5040 | 1.6275 | 0.5366 | 0.8320 | 0.5076 |
| Qwen3-4B (Equalized/R2G) | 0.5086 | 0.1844 | 0.3333 | 0.2000 | 0.4643 | 1.8075 | 0.6098 | 0.8640 | 0.5269 |
| Qwen3-4B (EM/R2G) | 0.5076 | 0.1417 | 0.3333 | 0.1576 | 0.4563 | 1.7375 | 0.6341 | 0.8640 | 0.5154 |
| Qwen3-4B (R2G/R2G) | 0.4629 | 0.1687 | 0.3974 | 0.1333 | 0.5794 | 1.5975 | 0.4634 | 0.8640 | 0.4895 |
| Qwen3-4B (Equalized/Sum) | 0.4456 | 0.1615 | 0.2308 | 0.1333 | 0.4524 | 1.7150 | 0.4878 | 0.8400 | 0.4656 |
| *Open-Source (Raw Model)* | | | | | | | | | |
| Qwen3-32B (Raw) | 0.1724 | 0.1510 | 0.1538 | 0.0000 | 0.4841 | 1.8300 | 0.5610 | 0.7920 | 0.3128 |
| Qwen3-14B (Raw) | 0.1924 | 0.1417 | 0.1667 | 0.1030 | 0.5317 | 1.7000 | 0.5854 | 0.5120 | 0.3027 |
| Qwen3-4B (Raw) | 0.1405 | 0.0854 | 0.0769 | 0.0364 | 0.4048 | 1.7400 | 0.4878 | 0.8560 | 0.2929 |
| *Closed-Source* | | | | | | | | | |
| Gemini-2.5-Pro | 0.3468 | 0.2740 | 0.4103 | 0.1939 | 0.4246 | 1.5900 | **0.9024** | **0.9280** | 0.4702 |
| Gemini-2.5-Flash | 0.2553 | 0.1958 | 0.3205 | 0.1212 | 0.4087 | 1.6850 | 0.6341 | **0.9280** | 0.3973 |
| GPT-4o | 0.3643 | **0.2917** | 0.2821 | 0.0303 | 0.3770 | 1.8975 | 0.8537 | 0.8800 | 0.4449 |
| GPT-4o-mini | 0.0976 | 0.0906 | 0.1538 | 0.2061 | 0.5317 | 0.2500 | 0.0488 | 0.3520 | 0.1729 |

Table 3: The main evaluation results on gym environments. For each gym, the highest performance within section is marked with underline, while the global best performance is highlighted in **bold**.

## 4.1 EXPERIMENT SETTINGS

**Settings.** We evaluate four representative configurations, denoted as A/B where A refers to the turn-level reward shaping method and B refers to the trajectory-level scoring method: Equalized/Sum, Equalized/R2G, EM/R2G, and R2G/R2G. The *Naive* setting is excluded because in practice its sparse effective reward signal, where many turns yield zero reward, quickly leads to training collapse. Among these variants, Equalized/Sum serves as the most natural extension of the original GRPO algorithm into the multi-turn setting, treating every turn equally while summing the incremental rewards to produce the trajectory score. From this baseline, we vary either the turn-level reward or the trajectory-level score to enable fair comparison across methods.

**Training.** All RL models are initialized with an SFT cold start, which stabilizes optimization and improves downstream RL performance; we analyze this effect in later section. We employ VERL (Sheng et al., 2024) for RL training. Each step samples a batch of 128 with 8 responses per query, training for 15 epochs in total (see Appendix E for full configuration). To encourage exploration, we remove KL regularization and use temperature 1.0. For all gyms, we employ Qwen3-32B as the simulated user model, set max turns to 16, and keep other settings as in Appendix C.

**Data.** We use TravelGym, TurtleGym, FunctionGym, TauGym, and PersuadeGym for training, while reserving IntentionGym, TelepathyGym, and SearchGym as held-out evaluation environments. This ensures test-time evaluation requires generalization to unseen interaction purposes. For the training split, we also retain in total 1K trajectories from the five training gyms as supervised fine-tuning data. These trajectories are distilled using GPT-4o as both agent and simulated user, providing consistent high-quality supervision for SFT initialization. All remaining trajectories are then used for RL training. Please refer to detailed statistics in Table 2.

**Model and Metrics.** We conduct experiments with Qwen3 models of 4B and 8B, and include larger variants (14B, 32B) for raw performance comparison. For closed-source baselines, we report GPT and Gemini families as references. Evaluation metrics follow each gym's definition: for TravelGym and TauGym, we adopt UserBench and Tau-Bench protocols and score by correctness of the final choice or state; for others, the score is the sum of turn rewards since each turn reflects incremental gain (equivalent to the metric definitions in Table 2). In addition to per-gym results, we report micro-averaged performance across all eight gyms for overall comparison.

## 4.2 EXPERIMENT RESULTS

We present our main results in Table 3. The key findings are presented in the following.

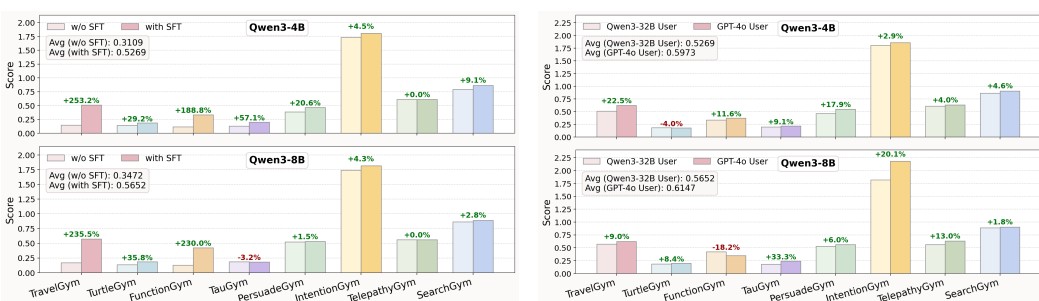

Figure 2: **Left:** SFT cold start improves RL training performance compared to direct RL on raw Qwen3 models. **Right:** GPT-4o as simulated user during training yields better downstream performance than Qwen3-32B as simulated user in gym environments.

**Equalized/R2G consistently outperforms other training settings.** The *Equalized/R2G* setting achieves the best performance across 4B and 8B models, while *Equalized/Sum* performs worst. This highlights the importance of trajectory-level score calculation: R2G outperforms simple summation because it better captures the cumulative value of intermediate steps toward reward-yielding turns. Importantly, zero reward does not imply zero contribution: for example, in *TelepathyGym*, asking clarifying questions earns no reward but helps narrow the answer space and thus supports final success. Assigning such turns zero advantage undermines learning, which explains the collapse observed under naive turn-level reward training. Finally, we find that the choice of turn-wise reward assignment (Equalized, EM, Distance-based) has less impact on performance and the simple Equalized scheme already suffices, suggesting trajectory-level scoring is more decisive than fine-grained turn differentiation. We further discuss this in Section 5.

**Gym-trained models can surpass closed-source ones in interactive tasks.** Our gym-trained models achieve higher overall performance than leading closed-source models, with Qwen3-8B notably outperforming Gemini-2.5-Pro and GPT-4o in *TravelGym*, *PersuadeGym*, and *IntentionGym*. These gains suggest that RL in our environments is particularly effective at enhancing direct user interaction and communication skills. However, closed-source models still dominate in environments such as *TelepathyGym* and *SearchGym*, which require integration with external tools (e.g. search engine) and strategic guessing. This contrast underscores that success in user-centric interaction is a deeply synthetic capability: it depends not only on refined reward shaping but also on strengthening broader competencies such as robust tool use and curiosity-driven reasoning.

**Scaling raw model size is less effective without interaction training.** We find that scaling raw Qwen3 model size yields only marginal gains in our evaluations: larger models show little advantage when they lack robustness in user-interaction abilities. By contrast, RL in our environments unlocks improvements brought by scaling: for 4B and 8B models trained under the best setting, their performance gap can surpass that of the raw 4B and 32B models. This shows scaling effects emerge most clearly after adaptation to user-centric interaction, suggesting foundational capacity alone is insufficient unless paired with training that elicits effective communication and user adaptability.

**Adapting existing benchmarks to user-centric settings can reduce performance.** In *TravelGym* and *TauGym*, we observe that models perform far worse than in the original UserBench (Qian et al., 2025b) and Tau-Bench (Yao et al., 2024), even with unchanged test data and metrics. Our only modification is the introduction of a standardized tool interface to mediate user interactions. This result suggests several insights: (1) prior results may partially reflect data leakage or overfitting to benchmark-specific patterns; (2) interacting correctly through standardized tools remains a significant challenge even for strong models; (3) user-centric abilities, such as structured communication and consistent tool use, are still not robust in current models, leaving room for improvement.

### 4.3 ANALYSIS

**Effectiveness of SFT cold start.** To validate our choice of using SFT cold start, we further trained raw Qwen3 4B and 8B models under the *Equalized/R2G* setting, which gave the best results in

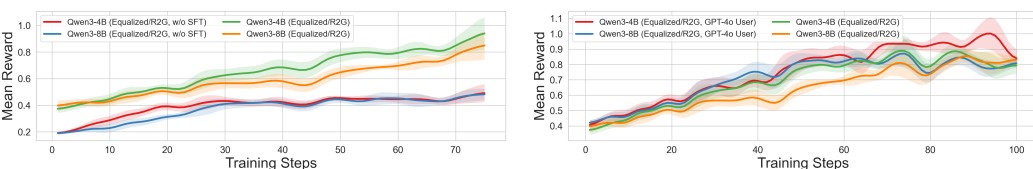

Figure 3: Comparisons of Qwen3 training curves under the Equalized/R2G setting: (1) **Left:** w/ vs. w/o SFT cold start and (2) **Right:** using GPT-4o vs. Qwen3-32B as the simulated user.

Table 3. As shown on the left of Figure 2, SFT cold start consistently allows RL to boost performance, sometimes even exceeding 100% gains. The training curves on the left of Figure 3 also show that SFT-initialized models begin from a stronger baseline and keep improving, while models without SFT plateau early at lower levels. Therefore, we conclude that SFT cold start equips the models with essential initial user interaction abilities, which RL can then better refine and extend.

**Choice of user simulation model for training.** In all the main experiments, we used Qwen3-32B as the simulated user for cost reasons, while evaluation always employed GPT-4o (detailed justifications in Appendix F). To probe the effect of simulator choice, we additionally trained Qwen3 4B and 8B with GPT-4o as the simulated user under *Equalized/R2G* setting. As shown on the right of Figure 2 and Figure 3, GPT-4o as simulated user generally yields higher performance, shows faster growth and higher plateaus (albeit subtly). This improvement likely arises from the alignment of interaction patterns in training and evaluation, which enables the model to adapt more effectively. Still, our main results confirm that budget-friendly simulators like Qwen3-32B transfer well to stronger users, while stronger simulated users can further strengthen robustness and user-centric abilities. We further discuss effective user simulation in detail in Section 5.

**Effectiveness and efficiency of user interaction.** cross all settings, we first measure the average effective turns for each evaluated gym, defined as the number of turns a model takes before obtaining its last non-zero turn-wise reward. This metric captures whether the model can meaningfully leverage interaction turns to achieve reward. As shown in the left column of Table 4, our trained models generally exploit more interaction turns, while raw Qwen3 models often stop gaining reward after just two or three turns. Nevertheless, the evaluation maximum is set to 16 turns, and even the best model achieves only 6.6463 effective turns on average, suggesting much room for improvement. Building on the interaction efficiency definition in Qian et al. (2025b), we further conduct weighted-timing analysis, where the reward at turn $i$ is scaled by a weight of $1/(i+1)$, and the final metric is the sum of all weighted rewards. This emphasizes early rewards, encouraging models to achieve gains quickly rather than delaying progress. As shown in the right column of Table 4, trained models consistently score higher,

| Model | Effective Turns[↑] | Time-Weighted Performance[↑] |
|---|---|---|
| Qwen3-8B (Equalized/R2G) | **6.6463** | **0.6516** |
| Qwen3-8B (EM/R2G) | 5.8792 | 0.5586 |
| Qwen3-8B (R2G/R2G) | 5.1050 | 0.6510 |
| Qwen3-8B (Equalized/Sum) | 6.1842 | 0.4530 |
| Qwen3-4B (Equalized/R2G) | 6.1307 | 0.6423 |
| Qwen3-4B (R2G/R2G) | 5.7317 | 0.6213 |
| Qwen3-4B (EM/R2G) | 5.6743 | 0.5355 |
| Qwen3-4B (Equalized/Sum) | 5.3881 | 0.5118 |
| Qwen3-32B (Raw) | 2.8079 | 0.2852 |
| Qwen3-14B (Raw) | 3.4129 | 0.3211 |
| Qwen3-4B (Raw) | 2.2545 | 0.2674 |
| Gemini-2.5-Pro | 5.7731 | 0.5263 |
| Gemini-2.5-Flash | 4.2465 | 0.4525 |
| GPT-4o | 3.4087 | 0.4024 |
| GPT-4o-mini | 1.9461 | 0.1614 |

Table 4: Gym-based RL trained models generally has more effective turns on average, while gaining non-zero reward more efficiently.

with *R2G*-based models outperforming those using *Sum* as the trajectory score. This validates that the *R2G* design not only improves effectiveness but also fosters efficiency by incentivizing earlier successes in interaction. We further elaborate on this issue of interaction efficiency in Section 5.

**Interaction with real users.** To evaluate the robustness of our trained models with human, we evaluated our best 4B and 8B models on TurtleGym and TelepathyGym with real users (detailed settings in Appendix F). Our results show a surprising pattern: our models achieve even higher performance with real users than GPT-4o simulated ones. We examine the interaction logs and find that, while GPT-4o user often responds with short signals (e.g., *"Yes, No or Maybe"*), human

users treat tasks more like cooperative games and tend to provide richer cues. For instance, in TelepathyGym, they offered hints such as *"The event kind of happens in the past, but not that far"*, which effectively guided models to refine their guesses. These findings highlight that agentic models benefit from being treated as collaborators, and our models show robustness and adaptability under such cooperative dynamics.

| Model | TurtleGym | TelepathyGym |
|---|---|---|
| Qwen3-4B (GPT-4o User) | 0.1844 | 0.6098 |
| Qwen3-4B (Real User) | $0.2952_{\uparrow 0.1108}$ | $0.7805_{\uparrow 0.1707}$ |
| Qwen3-8B (GPT-4o User) | 0.1854 | 0.5610 |
| Qwen3-8B (Real User) | $0.3127_{\uparrow 0.1273}$ | $0.7805_{\uparrow 0.2195}$ |

Table 5: Comparison of GPT-4o simulated and real user test results.

## 5 DISCUSSIONS

**Limits of turn-wise reward differentiation.** While trajectory-level scores are clearly beneficial, our results highlight the limits of existing turn-wise reward formulations: EM collapses all zero-reward turns into the same intermediate score, failing to distinguish productive from unproductive actions (e.g., insightful vs. irrelevant questions in *IntentionGym*), while R2G overemphasizes proximity to reward-yielding turns and misses cases where crucial progress occurs earlier. These heuristics show that simple turn-level reward differentiation can be misleading. Therefore, it is necessary to develop finer reward signals that capture both incremental gains and the contextual role of each turn. Such signals may need to be environment-specific, as the nature of useful intermediate steps varies across gyms. Thus, a universal strategy is unlikely to suffice, and instead future research should aim to design adaptive or learned reward shaping mechanisms that better capture turn-level utility while preserving the strengths of trajectory-level score calculation.

**Balance of rigor and flexibility in user simulation.** User simulation is essential for scaling RL training and enabling dynamic multi-turn rollouts. To make trained agentic models robust to diverse user behaviors, simulations must incorporate variation in responses, such as different strategies, tones, and communication styles. Future extensions of our framework could introduce richer user profiles in gym environments to better approximate real-world diversity and further strengthen robustness. At the same time, benchmarks must preserve rigor: when user interaction is the primary challenge, fairness across models is critical to ensure comparability. This inevitably reduces naturalness and requires rule-based structures to maintain consistency. In our eight gym designs, we address this tension by combining LLM-driven dynamic responses with rule-based strategies that track task completion. For example, in TravelGym the simulated user categorizes the agent's utterance before generating a guided response. This illustrates the inherent tradeoff between flexibility and rigor in user simulation, highlighting an important direction for future work on balancing the two for robust agent training.

**Balance of efficiency and effectiveness in user interaction.** In Section 4.3, we introduced two complementary metrics to evaluate user interaction: *Effective Turns* and *Time-Weighted Performance*. While higher values for both are desirable, we should note that inherently they may conflict. Models that achieve effectiveness by exploiting many turns risk creating tedious, inefficient conversations, while overly efficient agents may fail to fully capture user intent. Our metrics address this by accounting for not only the number of turns but also the rewards they deliver. This also mirrors real-world tradeoffs: users may tolerate a few clarifying questions, but prolonged or repetitive probing quickly becomes frustrating. Therefore, training and evaluation should consider how to balance efficiency and effectiveness, potentially through improved metrics and reward designs that encourage models to be both accurate and concise in user interaction.

## 6 CONCLUSION

In this paper, we introduced UserRL, a framework that provides eight distinct gym environments with a standardized interaction interface for training and evaluating user-centric agents. By leveraging turn-wise user feedback and rewards, we proposed customizable strategies for trajectory-level scoring and turn-level reward assignment, enabling systematic analysis of their impact on user-centric RL training. Looking ahead, we see several promising directions for extending UserRL: designing richer gym environments that balance rigor with flexibility, refining reward formulations that jointly capture effectiveness and efficiency, and exploring more diverse user simulation profiles. Altogether, we envision UserRL as a foundation for developing agentic models that act not merely as executors, but as user-centric collaborators capable of adapting to diverse needs in real-world interactions.

## ETHICS STATEMENT

Our study focuses on user-centric reinforcement learning within simulated gym environments and LLM-based user interactions. No personally identifiable data or sensitive human information was used; all user feedback in training derives from simulated or consenting human evaluators. We acknowledge potential risks, such as biases introduced by LLM simulators and misuse of trained conversational agents, and we mitigate these by releasing standardized, transparent environments and emphasizing safe, cooperative behaviors. Our research adheres to the ICLR Code of Ethics by prioritizing fairness, privacy, and research integrity.

## REPRODUCIBILITY STATEMENT

We have taken steps to ensure reproducibility of our findings. All gym environments, task definitions, and reward specifications are described in detail in the main paper (Section 2 and Section 4). Hyperparameters for GRPO training, SFT initialization, and evaluation setups (both simulated and real users) are also explicitly documented in the appendix (Appendix C and Appendix E). To facilitate replication, we provide our anonymized codebase in the supplementary material, covering environment implementations, model training scripts, and evaluation pipelines. Together, these materials allow other researchers to reproduce and extend our results with minimal ambiguity.

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

APPENDIX

# A  RELATED WORKS

**User-centric agent design and evaluation.**    As LLM agents become integrated into everyday use, research has focused on both evaluating and improving their alignment with complex, evolving user needs. On the evaluation front, benchmarks grounded in real user interactions capture underspecified or multi-intent queries and in-the-wild scenarios (Wang et al., 2024a; Qian et al., 2024b; Lin et al., 2024), while multi-turn testbeds probe an agent's ability to incorporate feedback, use tools, and adapt to shifting goals (Wang et al., 2024b; Yao et al., 2024; Barres et al., 2025). Personalized evaluation suites further explore whether models can infer and sustain user-specific preferences across long conversations (Zhao et al., 2025a). Complementing these, agent design advances teach LLMs to clarify ambiguity rather than hallucinate intent (Zhang et al., 2024; Chen et al., 2025a), and to personalize outputs via explicit or latent user modeling (Gao et al., 2024; Singh et al., 2024). Meanwhile, progress in reinforcement learning, such as MAU-RL, has begun to unlock more robust user-centric behaviors in agentic LLMs (Zhao et al., 2025b). Building on these directions, our work introduces UserRL, a unified framework that jointly benchmarks and trains agents on user-centric RL abilities within standardized gym environments.

**Agentic RL training and adaptation.**    Previous studies have primarily relied on supervised fine-tuning with carefully curated datasets to enhance LLMs' agentic abilities, such as tool use (Schick et al., 2023; Qin et al., 2024). More recently, reinforcement learning has emerged as a scalable and efficient training paradigm that fosters deeper reasoning and broader generalization. Early methods such as PPO (Schulman et al., 2017) and RLHF (Ouyang et al., 2022) laid the foundation, which has since evolved into preference-based algorithms including DPO (Rafailov et al., 2023), SimPO (Meng et al., 2024), and more recent trajectory-level approaches such as GRPO (Guo et al., 2025) and its variants DAPO (Yu et al., 2025) and VAPO (Yuan et al., 2025). These successive refinements improve training stability and efficiency, enabling RL to scale effectively to large models. In parallel, agentic RL frameworks extend beyond single-turn alignment to multi-turn interactions and long-horizon reasoning, including trajectory-level preference optimization (Shani et al., 2024), direct trajectory optimization via StarPO (Wang et al., 2025), and tool learning without step-level supervision (Singh et al., 2025). Such advances have broadened LLMs' agentic capabilities across diverse applications, including search (Jin et al., 2025), tool use (Qian et al., 2025a), and value judgment (Chen et al., 2025b). Furthermore, agentic RL has begun to intersect with unsupervised adaptation through test-time optimization (Zuo et al., 2025) and extend into other modalities such as vision–language modeling (Shen et al., 2025). Building on these, our work focuses on customizing RL for user-centric tasks, with particular emphasis on reward shaping at the turn and trajectory levels.

# B  NOVELTY AND CONTRIBUTIONS.

Although our framework builds upon established RL and simulator components, the novelty of this work lies in the way these elements are *systematically unified and decoupled* to enable controlled analyses of interactive, user-centric agents. Specifically, we contribute (i) a standardized *tool interface* across heterogeneous environments, which allows turn-level credit assignment to be studied in a principled and reproducible manner; (ii) a *decoupling of reward shaping and trajectory-level scoring*, which for the first time makes it possible to causally compare how shaping strategies interact with GRPO; (iii) a suite of *multi-turn user simulators in gym environments* that bridge SFT and RL training by exposing realistic patterns of ambiguity, clarification, and failure, providing a testbed for distributional robustness and transfer analyses; and (iv) a set of *evaluation metrics beyond raw task success*, including time-weighted efficiency in analysis and user-centric task completion in gyms, that expand the space of what is measurable for user-centric agents. We believe these contributions go beyond incremental extensions: they establish a new paradigm for probing the *mechanisms of alignment and credit assignment in user-centric RL*, positioning UserRL not only an engineering consolidation but foundation for theory-driven investigations of reward design and interactive generalization.

## C   GYM CONSTRUCTION DETAILS

**IntentionGym.**   IntentionGym is an environment designed to evaluate an agent's ability to uncover a user's true intent when given vague or underspecified tasks. The core task logic requires the agent to iteratively ask targeted clarifying questions until all critical missing details are revealed. Rewards are assigned based on the importance of the uncovered detail (high, medium, low), with penalties for unfocused or overly broad questions, encouraging efficient and precise inquiry. The dataset includes diverse user tasks across domains such as travel, education, and technology, each annotated with missing details and importance levels. The environment thus provides a principled testbed for intent understanding and ambiguity resolution through multi-round user–agent interaction.

The IntentionGym environment processes each action through a sophisticated two-step evaluation system. When an agent submits a question as input (corresponding to `Action` operation choice), the environment performs two parallel language model calls: (1) a response generation call that creates a natural conversational response from the simulated user's perspective without knowing the ground truth missing details (using temperature 0.7 for naturalness), and (2) an evaluation call that analyzes which specific missing details the question addresses (using temperature 0.0 for consistency). The evaluation model receives the complete list of remaining missing details with their importance levels (1=Low, 2=Medium, 3=High) and returns indices of covered details plus analysis metadata. The reward calculation uses a tiered system where High importance details yield 1.0 base reward, Medium yield 0.7, and Low yield 0.4, with a multi-detail penalty of 0.2 per additional detail covered (encouraging focused questions), multiplied by the configurable reward scale (default 1.0), then reduced by the step penalty (default 0.0), and finally normalized to [0,1] if configured. The environment tracks conversation history, updates covered/remaining detail lists, and terminates when all missing details are covered or maximum steps (default 20) is reached. The simulated user role is to respond authentically as someone seeking help while maintaining ignorance of what specific details need clarification, creating a realistic intention-guessing scenario where the agent must strategically ask questions to uncover the hidden missing information.

Please refer to Figure 4 and Figure 5 for system instruction details about the response generation and coverage evaluation processes.

**PersuadeGym.**   PersuadeGym is an environment designed to evaluate an agent's ability to persuade a user to change its stance on controversial statements through strategic argumentation. The core task logic requires the agent to iteratively present compelling arguments to move the user from an initial "Strongly Agree" position toward "Strongly Disagree" on various claims. Rewards are assigned based on the magnitude of stance change achieved, with exponential scaling that encourages significant persuasion rather than incremental shifts. The dataset includes diverse controversial statements across domains such as technology policy, social issues, and governance, each with detailed initial arguments supporting the the user's starting position. The environment thus provides a principled testbed for persuasive communication and strategic reasoning through multi-round adversarial dialogue.

The PersuadeGym environment processes each action through a stance evaluation system. When an agent submits a persuasive argument as input (corresponding to `Action` operation choice), the environment performs a single language model call that both generates a natural response and evaluates stance change (using temperature 0.0 for consistency). The evaluation model that simulates the user receives the current statement, the initial argument, conversation history, and the agent's latest argument, then returns a JSON response containing the feedback, updated stance, and internal reasoning. The stance progression follows seven discrete levels: Strongly Agree (0.0), Agree (0.167), Partly Agree (0.333), Neutral (0.5), Partly Disagree (0.667), Disagree (0.833), and Strongly Disagree (1.0). The reward calculation uses exponential scaling where the reward equals the number of levels changed toward disagreement divided by 6 (the maximum possible change), with no reward for moving back toward agreement or no change. The environment tracks conversation history, updates the simulated user's stance, and terminates when the user reaches "Strongly Disagree" (complete persuasion success) or maximum steps (default 20) is reached. The simulated user's role is to respond authentically as someone holding a strong initial position while being open to compelling evidence, creating a realistic persuasion scenario where the agent must strategically craft arguments to overcome resistance.

810
811
812
813
814
815
816
817
818
819
820
821
822
823
824
825
826
827
828
829
830
831
832
833
834
835
836
837
838
839
840
841

```
IntentionGym LLM Instruction for Response Generation

You are a person who has posted a vague request for help and is now
responding to someone who is trying to help clarify your needs.

Your job is to respond naturally as the person who originally made
the request.  Follow these guidelines:

1. If the question is asking about your specific preferences for this
task:
- Provide an authentic and coherent response
- Share realistic preferences that someone might have for this type of
task
- Be conversational and natural

2. If the question is NOT directly about your preferences for this
task:
- Try to answer helpfully if you can
- Guide the conversation back to clarifying what you need for your task
- Be polite but redirect:  "That's interesting, but what I'm really
trying to figure out is..."
- Do NOT provide what missing details need to be clarified or give any
examples.
- Do NOT provide concrete help or solutions - you're the one seeking
help!

Please respond in the following json format: {
    "thought":  "Your thought process about whether the question is
about your preferences and how to respond",
    "response":  "Your natural conversational response"
}

IMPORTANT:
- Respond only as the person seeking help, not as an evaluator
- Be natural and conversational
- Don't reveal any "ground truth" or act like you know what details are
missing
- Just respond authentically as someone who made this request
```

Figure 4: IntentionGym LLM system instruction for response generation.

Please refer to Figure 6 for system instruction details about the AI's response generation and stance evaluation process.

**TurtleGym.** TurtleGym is an environment designed to evaluate an agent's ability to engage in creative reasoning and contextual adaptation through interactive story-based puzzle solving with a user. The core task logic requires the agent to play a "turtle soup" game where they must uncover hidden twists in mysterious stories by asking strategic questions and providing comprehensive explanations. The environment presents the agent with a surface-level story description and challenges them to discover the underlying "bottom" truth through iterative inquiry and creative interpretation. Rewards are assigned based on how well the agent's final story explanation covers the evaluation criteria, with weighted scoring that emphasizes the most critical story elements. The dataset includes diverse mysterious scenarios with hidden twists, each annotated with evaluation criteria and importance weights. The environment thus provides a principled testbed for creative reasoning and contextual adaptation through multi-round collaborative story exploration.

The TurtleGym environment processes each action through a dual-mode evaluation system. When an agent submits an inquiry action (corresponding to `Action` operation choice), the environment performs a single language model call that evaluates whether the question is helpful for understanding the story, returning "Yes", "No", or "Maybe" responses (using temperature 0.0 for consistency). When an agent submits an answer action (corresponding to `Answer` operation choice), the environment evaluates the story explanation against multiple weighted criteria, where each criterion receives a

---

**IntentionGym LLM Instruction for Coverage Evaluation**

```
You are evaluating how well a user's question addresses missing details
in a vague task.

Your job is to evaluate which (if any) of the missing details are
addressed by the latest question.

Rules for evaluation:
- If the question is NOT related to clarifying the user's intent or
task requirements, covered_detail_indices should be an empty list
- For each missing detail that is directly addressed by the question,
note its index
- A question "addresses" a detail if it would help reveal the
information needed for that detail

Please respond in the following json format:
{
    "analysis":  "Brief explanation of what missing details (if any)
were covered by this question",
    "is_task_related":  true/false (whether the question is about
clarifying the task requirements),
    "covered_detail_indices":  [list of indices from the missing details
list that this question addresses]
}

IMPORTANT: You are ONLY evaluating, not generating responses.  Focus
solely on which details are addressed.
```

Figure 5: IntentionGym LLM system instruction for coverage evaluation.

---

**PersuadeGym LLM Instruction for Response Generation and Stance Evaluation**

```
## Task
You are participating in a persuasion conversation.  You hold a
position on a statement, and someone is trying to persuade you to
change your mind.  For each of their arguments, you should:

1. Respond naturally and thoughtfully to their argument
2. Express your current stance after considering their point
3. You may change your mind if presented with compelling evidence
4. Be respectful but critical - point out logical fallacies if you see
them

Follow the format strictly:
```json
{
    "thought":  "Your internal reasoning about their argument",
    "response":  "Your public response to their argument (max 200
tokens)",
    "stance":  "One of:  Strongly Agree, Agree, Partly Agree, Neutral,
Partly Disagree, Disagree, Strongly Disagree"
}
```
```

Figure 6: PersuadeGym LLM system instruction for response generation and stance evaluation.

score of 0.0 (completely incorrect), 0.5 (partially correct), or 1.0 (completely correct), multiplied by the criterion's weight to produce a final score between 0.0 and 1.0. The reward calculation uses incremental improvement: only answers that exceed the current best score receive positive rewards equal to the score improvement multiplied by the reward scale (default 1.0), with step penalties (default 0.0) applied cumulatively. The environment tracks the best score achieved and terminates when an answer reaches the success threshold (default 0.9) or maximum steps (default 20)

---

**TurtleGym LLM Instruction for Inquiry Evaluation**

```
## Task
You are a helpful assistant to respond to the user query based on the
given story scenario (surface) and ground truth (bottom) in a Turtle
Soup game.  Please follow the instructions below.

## Instructions
1. You can only give three values:  "Yes", "No", or "Maybe" in your
response.
2. "Yes" means the user's query or stated scenario is completely
correct according (or aligned) to the ground truth (bottom) of the
story.
3. "No" means the user's query is incorrect or contradicts the ground
truth (bottom) of the story, or the user's query is not even close to
the ground truth.
4. "Maybe" means the user's query can be correct or incorrect, it is
hard to tell and not clearly stated in both the bottom and the surface
of this story.  "Maybe" is usually used when the user's query is not
quite relevant to the ground truth.  Please try to be determinant and
use as less "Maybe" in your response as possible.

## Example Format

### Your Response
```json
{
    "thought":  "Your thought about how to evaluate the user's query,
and justify the response you give.",
    "response":  "Yes" or "No" or "Maybe"
}
```
```

Figure 7: TurtleGym LLM system instruction for inquiry evaluation.

is reached. The simulated user's role is to provide objective evaluation of the agent's questions and story explanations without revealing the ground truth, creating a realistic collaborative puzzle-solving scenario where the agent must strategically explore and creatively interpret mysterious scenarios.

Please refer to Figure 7 and Figure 8 for system instruction details about the inquiry evaluation and story explanation scoring processes.

**TelepathyGym.** TelepathyGym is an environment designed to evaluate an agent's ability to engage in strategic reasoning and hypothesis testing through interactive mind reading games with a user. The core task logic requires the agent to guess what entity the user is thinking of by asking strategic yes/no questions and making final guesses. The environment presents the agent with a category description and challenges them to systematically narrow down the possibilities through binary questioning, building a hypothesis space that converges on the correct answer. Rewards are assigned based on the accuracy of the final guess, with binary scoring that emphasizes successful entity identification. The dataset includes diverse entities across categories such as famous people, animals, objects, and landmarks, each with detailed descriptions for the simulated user's knowledge base. The environment thus provides a principled testbed for strategic reasoning and hypothesis testing through multi-round interactive deduction.

The TelepathyGym environment processes each action through a dual-mode evaluation system. When an agent submits an inquiry action (corresponding to `Action` operation choice), the environment performs a single language model call that evaluates the question against the target entity and returns "Yes", "No", or "Maybe" responses (using temperature 0.0 for consistency). When an agent submits an answer action (corresponding to `Answer` operation choice), the environment evaluates whether the final guess correctly identifies the target entity, returning a binary score of 1.0 for exact matches or 0.0 for incorrect guesses. The final reward is calculated by multiplying the reward scale (default 1.0), with step penalties (default 0.0) applied cumulatively. The environment tracks clue history from

---

**TurtleGym LLM Instruction for Inquiry Evaluation**

```
## Task
You are a helpful agent to help me evaluate the correctness of the
user's story against the ground truth in a Turtle Soup game.  You
should give both your score and evaluation feedback based on a
evaluation protocol provided.  Please follow the instructions below.

## Instructions
1. There may exist multiple evaluation criteria based on the evaluation
protocol.  You should give a score for each criteria.
2. Your score can only take three values:  0, 0.5, 1.0, where 0 means
the user's answer is completely incorrect (not even close to the ground
truth), 0.5 means the user's answer partially aligns with the ground
truth, and 1.0 means the user's answer is completely correct.
3. After giving the score, you should give an overall feedback about
which part in the user's answer is correct (or the story is all wrong
and totally not aligned).  Do not say which part is incorrect or not
aligned with the ground truth.  Do not release anything else about the
ground truth (bottom) or the evaluation protocol.  Try to keep your
feedback concise and to the point.

## Example Format

### Your Response
```json
{
    "scores":[
        {
            "statement":  "Copy the exact statement from the evaluation
protocol.",
            "thought":  "Your thought about how to evaluate the
statement, and justify the score you will give based on the protocol
statement and comparison between the ground truth and user's answer.",
            "score":  0 or 0.5 or 1.0
        },
    ...  (the number of scores should be the same as the number of
criteria in the evaluation protocol, and the order should also exactly
match)
    ],
    "feedback":  "Your feedback to the user's answer about which part
is correct.  Do not release anything about the ground truth (bottom)
and the evaluation protocol.  Be concise and to the point.  Use the
second person tone (you / your) to address the user."
}
```
```

Figure 8: TurtleGym LLM system instruction for story explanation scoring.

previous questions and responses, updates the best score achieved, and terminates when a correct guess is made (score 1.0) or maximum steps (default 20) is reached. The simulated user's role is to respond honestly to yes/no questions based on the target entity while maintaining the mystery, creating a realistic mind reading scenario where the agent must strategically formulate questions to systematically eliminate possibilities and converge on the correct answer.

Please refer to Figure 9 and Figure 10 for system instruction details about the question response and entity guessing evaluation processes.

**FunctionGym.** FunctionGym is an environment designed to evaluate an agent's ability to engage in mathematical reasoning and pattern generalization through interactive function discovery with a user. The core task logic requires the agent to uncover a hidden mathematical mapping rule by testing different number combinations and analyzing the results to identify the underlying pattern. The environment presents the agent with a four input numbers and challenges them to discover the

```
TelepathyGym LLM Instruction for Question Response

## Task
You are a telepathic entity playing a mind reading game.  The user
is trying to guess what entity you are thinking of by asking yes/no
questions.  You should respond honestly based on the target entity
you're thinking of.

## Instructions
1. You are thinking of a specific entity (person, object, concept,
etc.)  – this is the "target_entity" provided to you.
2. The user will ask questions to narrow down what you're thinking of.
3. Answer "Yes" if the question is true about your target entity.
4. Answer "No" if the question is false about your target entity.
5. Answer "Maybe" only if the question is ambiguous or you genuinely
cannot determine a clear yes/no answer.
6. Be helpful and honest – the goal is for them to eventually guess
correctly through good questions.

## Example Format

### Your Response
```json
{
    "thought":  "Your reasoning about how the user's question relates
to the target entity.",
    "response":  "Yes" or "No" or "Maybe"
}
```
```

Figure 9: TelepathyGym LLM system instruction for question response.

mapping function through systematic experimentation and logical deduction. Rewards are assigned based on the accuracy of the final answer for a test case, with binary scoring that emphasizes successful mathematical reasoning. The dataset includes diverse mathematical functions with varying complexity levels, each containing a hidden mapping rule, test case, and expected result. The environment thus provides a principled testbed for mathematical reasoning and pattern generalization through multi-step interactive problem solving.

The FunctionGym environment processes each action through a three-mode evaluation system. When an agent submits a four input numbers (corresponding to Action operation choice), the user evaluates them by applying the hidden function rule and returns the calculated result, with no reward for calculation actions. When an agent submits a search action (corresponding to Search operation choice), the user retrieves the test case numbers, providing the agent with the specific input values for the final evaluation. When an agent submits an answer action (corresponding to Answer operation choice), the environment evaluates whether the numerical answer matches the expected result within a tolerance of 1e-6, returning a reward of 1.0 for correct answers or 0.0 for incorrect ones. The final reward is calculated by applying the correct answer reward (default 1.0) or incorrect answer reward (default 0.0), with step penalties (default 0.0) applied cumulatively. The environment tracks action history, updates the answer status, and terminates when a correct answer is submitted or maximum steps (default 20) is reached. The simulated user's role is to provide accurate mathematical calculations based on the hidden rule while maintaining the mystery of the underlying mapping function, creating a math reasoning scenario where the agent must systematically test hypotheses and generalize from observed patterns to discover the correct mapping rule.

All the user response in this gym is rule-based so there is no LLM simulation for user specifically in this gym.

**SearchGym.** SearchGym is an environment designed to evaluate an agent's ability to demonstrate general helpfulness and reasoning through interactive web search and question answering with a user. The core task logic requires the agent to answer general knowledge questions from the user by performing web searches to retrieve relevant information and synthesizing the results into accurate

1080
1081
1082
1083
1084
1085
1086
1087
1088
1089
1090
1091
1092
1093
1094
1095
1096
1097
1098
1099
1100
1101
1102
1103
1104
1105
1106
1107
1108
1109
1110
1111
1112
1113
1114
1115
1116
1117
1118
1119
1120
1121
1122
1123
1124
1125
1126
1127
1128
1129
1130
1131
1132
1133

### TelepathyGym LLM Instruction for Entity Guessing Evaluation

```
## Task
You are a telepathic entity playing a mind reading game.  The user is
trying to guess what entity you are thinking of based on the clues
you've given through your "Yes" or "No" responses to their questions.
You need to evaluate if their final guess is correct.

## Instructions
1. You are thinking of a specific entity (person, object, concept,
etc.)  – this is the "target_entity" provided to you.
2. The user has been asking questions about this entity and is now
making a final guess.
3. You should evaluate if their guess correctly identifies the target
entity you were thinking of.
4. Only return "Yes" if their guess is exactly correct or a clearly
equivalent/synonymous identification of the target entity.
5. Return "No" if their guess is wrong, partially correct, or close but
not exact.
6. There is NO partial credit – it's either completely right (Yes) or
wrong (No).
7. Address the user in second person tone (e.g., "You", "Your",
"You're") in your feedback.
8. Your feedback should be concise and do not release anything
about the target entity.  Just state your judgment and encourage or
congratulate the user.

## Example Format

### Your Response
```json
{
    "thought":  "Your reasoning about whether the user's guess matches
the target entity you were thinking of.",
    "judgment":  "Yes" or "No",
    "feedback":  "Brief feedback explaining why their guess is correct
or incorrect.  Do not reveal the correct answer if they are wrong."
}
```
```

Figure 10: TelepathyGym LLM system instruction for entity guessing evaluation.

responses. The environment presents the agent with diverse questions across various domains and challenges them to leverage online information sources effectively to provide correct answers. Rewards are assigned based on the accuracy of the final answer, with binary scoring that emphasizes successful information retrieval and synthesis. The dataset includes diverse general knowledge questions from the Bamboogle benchmark, each requiring multi-step reasoning and information gathering. The environment thus provides a principled testbed for general helpfulness and reasoning through multi-step search-based question answering.

The SearchGym environment processes each action through a dual-mode evaluation system. When an agent submits a search action (corresponding to `Search` operation choice), the environment performs a web search using the Serper API and returns formatted search results with titles and snippets, with no reward for search actions but a limit on maximum search steps (default 5). When an agent submits an answer action (corresponding to `Answer` operation choice), the environment evaluates the answer using either rule-based comparison (normalized string matching) or LLM-based evaluation (using temperature 0.0 for consistency, default method), returning a reward of 1.0 for correct answers or 0.0 for incorrect ones. The final reward is calculated by applying the correct answer reward (default 1.0) or incorrect answer reward (default 0.0), with step penalties (default 0.0) applied cumulatively. The environment tracks search history, updates the answer status, and terminates when a correct answer is submitted or maximum steps (default 20) is reached. The simulated user's role is to provide accurate web search results by interacting with the real web backend and evaluate answer correctness while maintaining the challenge of information synthesis, creating a search-based question answering

SearchGym LLM Instruction for Answer Evaluation

```
## Task
You are asked to judge whether the answer for a question is correct or
not.

## Instructions
1. You will be provided with the question, the model's answer, and the
correct answer.
2. If the answer is exactly the same, or a clearly
equivalent/synonymous identification of the correct answer, return
"Yes".  Please base your answer judgment on the given question
scenario, instead of just comparing the answers.
3. If the answer is wrong, return "No".
4. In your feedback, you could provide a succinct explanation for your
judgment, but you should never reveal the correct answer.
5. In your feedback, please use second person tone (e.g., "You",
"Your", "You're").
6. In your feedback please do not give any hint or any information
about the correct answer.

## Example Format

### Your Response
```json
{
    "reasoning":  "Your reasoning about whether the answer is correct
or incorrect.",
    "judgment":  "Yes" or "No",
    "feedback":  "Brief feedback explaining why the answer is correct
or incorrect.  Do not reveal the correct answer if they are wrong."
}
```
```

Figure 11: SearchGym LLM system instruction for answer evaluation.

scenario where the agent must strategically formulate search queries and synthesize information from multiple sources to provide accurate responses.

Please refer to Figure 11 for system instruction details about answer evaluation processes.

**TauGym.** TauGym is an environment designed to evaluate an agent's ability to demonstrate tool use and task-oriented interaction through comprehensive user requirement fulfillment with a simulated user. The core task logic requires the agent to accomplish complex user requests by leveraging available internal tools, gathering necessary information through conversation, and executing appropriate tool calls to complete the specified objectives. The environment presents the agent with realistic user scenarios from retail and airline domains and challenges them to navigate multi-step workflows involving tool discovery, user communication, and systematic task execution. Rewards are assigned based on the original Tau-Bench implementation. The dataset includes diverse task-oriented scenarios from the existing Tau-Bench benchmark, each containing detailed user requirements and available tool sets. The environment thus provides a principled testbed for tool use and task-oriented interaction through multi-step problem solving.

The TauGym environment processes each action through a three-mode evaluation system. When an agent submits an interaction action (corresponding to `Action` operation choice), the environment facilitates direct communication with the simulated user and returns conversational responses. When an agent submits a search action (corresponding to `Search` operation choice), the environment retrieves available internal tools information or help documentation, providing the agent with comprehensive tool specifications and usage guidelines. When an agent submits an answer action (corresponding to `Answer` operation choice), the environment parses and executes tool calls using the existing Tau-bench framework, returning tool execution results and task completion status. The final reward is

calculated by the tau-bench evaluation system for each step based on task completion quality and user satisfaction, with additional step penalties (default 0.0) applied cumulatively. The environment tracks action history, maintains tool information, and terminates when the task is successfully completed or maximum steps (default 30) is reached. The simulated user's role is to provide realistic task requirements and respond authentically to agent interactions while maintaining the complexity of task-oriented scenarios, creating a comprehensive evaluation environment where the agent must strategically combine conversation, tool discovery, and systematic execution to fulfill user requirements effectively.

All the user responses are based on the implementation of original Tau-Bench so there is no additional LLM prompt or instruction introduced in this specifically adapted gym environment.

**TravelGym.** TravelGym is an environment designed to evaluate an agent's ability to demonstrate preference elicitation and personalized planning through comprehensive travel booking assistance with a user. The core task logic requires the agent to help users make personalized travel arrangements by eliciting their preferences across multiple dimensions (flight, hotel, rental car, apartment, restaurant) and providing tailored recommendations. The environment presents the agent with diverse travel scenarios and challenges them to systematically gather user preferences, search for relevant options, and make optimal choices that align with the user's needs. The dataset includes complex travel planning scenarios with multiple preference dimensions, each containing correct, wrong, and noise options to test the agent's ability to distinguish quality recommendations. The environment thus provides a principled testbed for preference elicitation and personalized recommendation through multi-round user-agent interaction.

The TravelGym environment processes each action through a three-mode evaluation system. When an agent submits an action (corresponding to `Action` operation choice), the environment performs a single LM call that evaluates whether the agent's utterance relates to preference elicitation, returning type classifications (1: normal conversation, 2: preference-related, 3: unavailable preference, 4: too vague) with corresponding rewards of 0.0, 0.2, 0.0, and 0.0 respectively (using temperature 0.0 for consistency). When an agent submits a search action (corresponding to `Search` operation choice), the environment evaluates search request alignment and returns travel options for valid dimensions, with successful searches yielding 0.2 reward and system errors simulated every 5 search attempts. When an agent submits an answer action (corresponding to `Answer` operation choice), the environment evaluates option selections against ground truth, with best options yielding 1.0 reward, correct but not the best options yielding 0.8 reward, and wrong choices incurring penalties (default 0.0). The final reward is calculated by multiplying the reward scale (default 1.0), with step penalties (default 0.0) applied cumulatively. The environment tracks conversation history, preference elicitation progress, and remaining options, terminating when all best options are selected or maximum steps (default 20) is reached. The simulated user's role is to respond authentically to preference questions while maintaining realistic travel planning constraints, creating a comprehensive scenario where the agent must strategically balance information gathering, search execution, and recommendation quality.

All the user responses are based on the implementation of UserBench, so there is no additional LLM prompt or instruction introduced in this specifically adapted gym environment.

# D  CLARIFICATIONS AND JUSTIFICATIONS

**Fidelity of User Simulator.** As discussed in Section 5, designing user simulators requires balancing flexibility and task rigor. To maintain naturalness while ensuring reproducibility, our LLM-based users are not entirely free-form but are guided by task instructions and ground-truth annotations. Conceptually, the user can be viewed as an "oracle with constraints": it classifies the agent's intent, evaluates responses against the ground truth, and produces replies according to carefully crafted instruction templates. This design ensures that feedback remains consistent, interpretable, and task-specific rather than arbitrary. To prevent trivial solution strategies (e.g., direct leakage of answers), certain gyms deliberately restrict the user's accessible information. We manually audited interactions under GPT-4o and Qwen3-32B that we employed in main experiments, including adversarial prompts, and found no evidence of answer leakage or instability. Empirically, the simulators consistently generated faithful and robust responses, which supports their reliability for rigorous training and

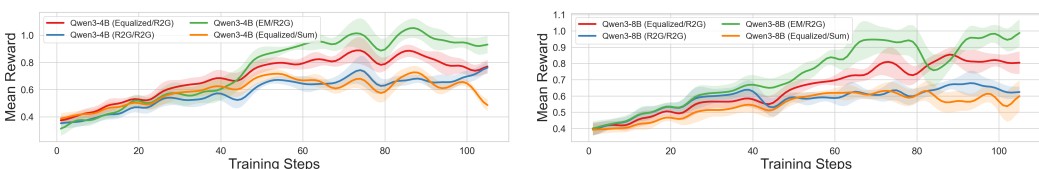

Figure 12: Training curves of Qwen3 4B (left) and 8B (right) models across different reward strategies.

evaluation. In summary, the deliberate instruction design of each gym ensures robustness without sacrificing reproducibility, a critical property for benchmarking user-centric agents.

**Parameter Selection and Sensitivity.** Our parameter settings are motivated by empirical tuning rather than arbitrary choice. For reward-to-go (R2G), we selected $\gamma = 0.8$ based on a grid search over $\{0.6, 0.7, 0.8, 0.9\}$ using Qwen3-4B. Values $\leq 0.5$ caused premature decay of future rewards and unstable training, while $\gamma = 1$ reduced to raw summation, leading to overemphasis on the first turn. $\gamma = 0.8$ yielded the best overall micro-average across gyms, balancing stability and forward credit propagation. For the Exponential Mapping (EM), we experimented with $k \in \{0.5, 1, 2, 3\}$. $k = 0.5$ consistently underperformed, whereas $k = 1, 2, 3$ produced nearly identical results (variance $< 1\%$), indicating robustness of the mapping curve within this range. We chose $k = 2$ as a representative setting that offers a sharper but stable reward gradient. These choices were thus informed by both principled considerations and empirical validation.

**Trajectory and Turn-Level Reward Shaping.** We selected our shaping strategies to cover both theoretically grounded and practically intuitive variants. R2G follows classic formulations of temporal credit assignment (Tamar et al., 2016), rewarding actions in proportion to their downstream impact and thereby encouraging efficiency in multi-turn interactions. EM, on the other hand, applies a monotonic convex transformation that emphasizes decisive turns by amplifying informative signals while compressing noise, reflecting the practical dynamics of dialogue where early clarifications are disproportionately valuable. We also included "baseline" strategies such as naive, equalized, and summation, which serve as natural points of comparison and reveal how much gain arises from more structured shaping. Together, these strategies form a principled yet diverse spectrum for probing credit assignment in user-centric environments. While our work highlights the effectiveness of R2G and EM, our open-source framework makes it straightforward to extend this analysis with additional shaping functions, ensuring that our method can evolve alongside future research.

**On External Benchmark Transfer.** We have not include additional external benchmarks beyond those already integrated into our gym suite for several reasons. First, several widely used user-centric benchmarks such as UserBench, IN3, and Tau-Bench are already instantiated within our framework as gyms, ensuring direct comparability without redundant evaluation. Second, our emphasis is on interactive agents operating through standardized tool-use abstractions, which many existing benchmarks do not natively support; adapting them faithfully would require significant re-engineering and could compromise comparability. Third, our main experiments already span a broad spectrum of capabilities, including reasoning, planning, retrieval, and dialogue grounding, with three gyms explicitly held out as unseen test environments in the main results table. This design allows us to study transfer and generalization without requiring every possible benchmark. Moreover, evaluating on heterogeneous external datasets often introduces confounding differences in annotation quality, interface format, or reward granularity, which may obscure rather than clarify the effects of reward shaping and trajectory scoring. Finally, we view further adaptation of external resources into our gym environment as a natural avenue for community-driven extension, which our open-source release is designed to facilitate. In this way, we balance rigor, breadth, and clarity of evaluation, while keeping the benchmark extensible for future work.

# E  TRAINING EXPERIMENT DETAILS

**Training Data.** As described in Section 3, we use 1k samples for SFT cold start before RL training. To construct this dataset, we collect all training tasks from five environments, where GPT-4o acts as

| RL Training Config | | |
|---|---|---|
| Parameter | Value | Notes |
| algorithm.gamma | 0.8 | Discount factor |
| algorithm.k | 2.0 | Scaling coefficient |
| data.train_batch_size | 128 | Training batch size |
| data.max_prompt_length | 1152 | Max input length |
| data.max_response_length | 8192 | Max output length |
| actor.optim.lr | 1e-6 | Learning rate |
| actor.ppo_mini_batch_size | 16 | PPO minibatch size |
| actor.use_kl_loss | False | KL loss disabled |
| actor.entropy_coeff | 0 | Entropy coefficient |
| rollout.name | sglang | Rollout engine |
| rollout.n | 8 | Parallel rollouts |
| rollout.multi_turn.max_turns | 16 | Multi-turn limit |
| trainer.n_gpus_per_node | 8 | GPUs per node |
| trainer.nnodes | 1 | Number of nodes |
| trainer.total_epochs | 15 | Training epochs |

| SFT Cold Start Config | | |
|---|---|---|
| Parameter | Value | Notes |
| finetuning_type | full | Full parameter tuning |
| cutoff_len | 16384 | Max input length |
| per_device_train_batch_size | 2 | Batch size per GPU |
| gradient_accumulation_steps | 4 | Grad accumulation |
| learning_rate | 1.0e-5 | Base LR |
| num_train_epochs | 3.0 | Training epochs |
| lr_scheduler_type | cosine | Scheduler type |
| warmup_ratio | 0.1 | Warmup fraction |
| bf16 | true | Mixed precision |

Table 6: RL and SFT Training Configurations.

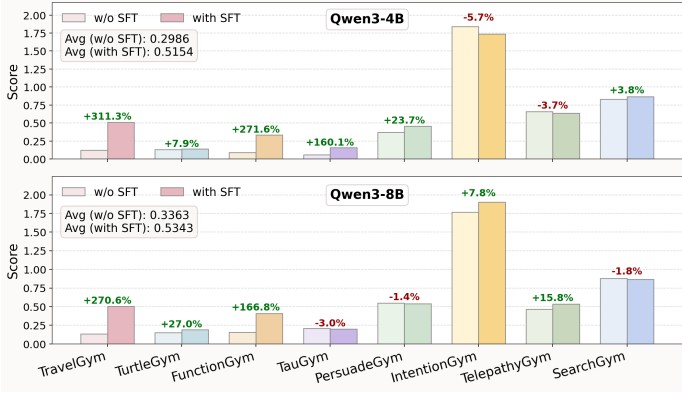

Figure 13: Additional experiment to validate SFT cold start improves RL training performance compared to direct RL on raw Qwen3 models.

both the simulated user and the agent model interacting with each gym. We then rank all interaction trajectories within each environment. For each environment, we select the top-$K$ ranked trajectories and combine them to form the final 1k-sample dataset used for SFT training. Beyond these 1k samples, all remaining training tasks are used for RL training. Since RL requires the actor model to roll out multiple trajectories, only task descriptions are needed rather than distilled trajectories.

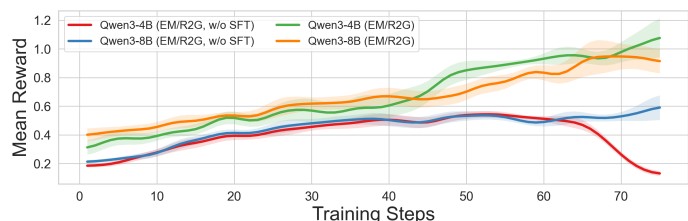

Figure 14: Additional experiment on comparisons of the training curves under the EM/R2G setting: w/ vs. w/o SFT cold start.

**Training Configuration.** The detailed training configurations are listed in Table 6. For SFT, we use 4 Nvidia H200 GPUs, and each training takes approximately 1 hour. For RL training, we use 8 Nvidia H200 GPUs, and each training takes approximately 1.5 days. We use LlamaFactory (Zheng et al., 2024) for SFT training and VERL (Sheng et al., 2024) for RL training. Especially for RL, we further reserve 5% of data as validation set and pick the best saved checkpoint for final result evaluation for each setting.

**Evaluation Configuration.** For evaluation, we keep all the model's temperature to 0.0 to ensure consistency. The maximum interaction turn is set to 16, while all other settings are set to default aligning with the description in Appendix C.

**Instructions and Tools.** We present the system prompt template for the agent in Figure 16. The content inside the placeholders varies depending on the gym that the agent interacts with; in the figure, we show an example from TauGym's system prompt. All tasks within the same gym share the same system prompt. Importantly, both training and evaluation for each gym use the same system prompt to ensure consistency. In addition, we provide the YAML schema of the `interact` tool, which defines the interface for agent-gym interaction, in Figure 15.

# F   ANALYSIS DETAILS

**Details of Training Curves.** For all curves in Figure 3, we report rewards using a consistent standard: the sum of turn-level rewards per trajectory. This ensures comparability across methods, even though the actual training signals may differ (e.g., R2G computes trajectory scores differently, and GRPO further transforms them into token-level advantages). Thus, the plotted rewards should be interpreted as a normalized measure for comparison rather than the raw training signal.

Training curves (solid lines) are smoothed with a Gaussian 1D filter ($\sigma = 2$). The shaded regions indicate fluctuations during training, defined as the deviation between the raw and smoothed values at each step. Their boundaries are further smoothed with the same filter for clarity.

**Justification for User Simulation.** We employ Qwen3-32B for user simulation during training due to the large number of requests involved. Specifically, the total number of requests in one training run is approximately: data size (2k) $\times$ epochs (15) $\times$ rollouts (8) $\times$ maximum turns per rollout (16) $\approx$ 4M requests. Using a closed-source model for such scale would be prohibitively expensive, making an open-source model a more budget-friendly choice for training.

During inference, however, the total number of requests is much smaller. Therefore, we use GPT-4o for user simulation at test time, as it provides stronger instruction-following and more realistic user behavior. This setup also allows us to verify whether a model trained with weaker user simulations can generalize effectively to interactions with a stronger simulated user during final evaluation.

**Real user participation.** For real user testing, we recruited five PhD students in computer science to serve as users behind TelepathyGym and TurtleGym. Participation was voluntary, with no payment involved, and all participants consented to the use of their interaction data for research purposes. Prior to the experiment, users were instructed on the purpose of the gyms, the evaluation procedure, and how to interact with the tested models. In both gyms, their primary role was to act as oracles,

```
YAML Tool Schema: interact_with_env
type: "function"
function:
    name: "interact_with_env"
    description: "A tool for interact with a target environment.  The
detailed environment description and action space is provided in the
system prompt, so please follow the system prompt when calling this
tool.  You can use this tool to interact with the target environment
step by step."
    parameters:
        type: "object"
        properties:
            choice:
                type: "string"
                enum: ["action", "answer", "search"]
                description: "Your choice of what to do next, must be one
of action, answer or search.  Please follow system prompt about the
scope of choices you can make and how to decide your choice."
            content:
                type: "string"
                description: "The content of your choice, must be a
string.  If you choose action, you should provide the action you want
to take.  If you choose answer, you should provide the answer that
you want to submit.  If you choose search, you should provide the
search query.  The specific format of the content is determined by
the environment description in the system prompt.  Please follow the
format strictly in order to successfully use this tool."
                required: ["choice", "content"]
```

Figure 15: Interact tool's schema for environment interaction used in all the training experiments. This tools serves as the standardized interface for agent's interaction with multiple gym environments.

judging the model's questions and answers against the ground truth, while ensuring they did not directly reveal the correct answers.

The five users divided all tasks evenly across both gyms. We emphasize that this is only a preliminary test: user behaviors naturally vary across individuals, so results may differ if other participants are employed. This issue of balancing rigor and flexibility is discussed further in Section 5. As observed in Section 4.3, models tended to perform better when interacting with real users. This improvement arises because, although users were instructed not to leak answers, they sometimes offered subtle hints rather than simply judging responses. We interpret this as evidence that real users instinctively treat agentic models as collaborators rather than executors, an insight that further underscores the significance of our work.

**Additional experimental results.**   For the main results, we provide the training curve regarding different turn-level rewarding strategies in Figure 12. We also present supplementary analysis of training raw Qwen3 4B and 8B models under the *EM/R2G* setting. The corresponding results and training curves are shown in Figure 13 and Figure 14. These findings further validate that models initialized with SFT cold start consistently outperform those trained without it, even under different reward shaping strategies. This strengthens our main claim that SFT initialization is critical for unlocking and sustaining effective user-centric RL training.

# G   USE OF LLMS

In this work, LLMs are used strictly for research support rather than as sources of substantive content. Their use falls into three categories: (i) serving as simulated users in our gym environments, (ii) providing baselines for evaluating trained models, and (iii) assisting with language refinement during paper writing. Both open-source and closed-source LLMs are employed for simulation and evaluation to ensure a comprehensive analysis. For writing support, we used GPT-5 solely to polish text (improving coherence and grammar) while all ideas, logic, results, and technical contributions

originate from the authors. To safeguard rigor, we have carefully reviewed all LLM-refined texts to confirm that no hallucinated content was introduced and that the original arguments, findings, and perspectives were faithfully preserved.

---

**Agent System Prompt Template (TauGym as Example)**

```
## Task
You are an agent that actively interact with a specific environment.
The following are the details of the environment and your action space.

## Environment Description
{{Placeholder:  The environment description for this gym; Example of
TauGym in the following}}

TauGym is an environment where you need to interact with both the
user and internal tools to fulfill the user's request.  You should
thoroughly understand the user's goal, figure out what information
is needed, and get this information through querying the user or
leveraging the internal tool step by step.

## Action Space
You should call the tool interact_with_env to interact with the
environment.  The action should be one of the following:  search,
action, or answer.

## Action Description
{{Placeholder:  The definition of particular action under this specific
gym's context; Example of TauGym in the following}}

   * search:  If you choose search, you must specify either 'tools'
or 'help' in the content field.  Giving 'tools' will return a list of
internal tools, including their descriptions and required arguments,
which you can later call through choosing answer.  Giving 'help' will
return a general guidance on how to interact with the environment
effectively.
   * action:  If you choose action, you will communicate directly with
the user through the message you write in the content field.  Ask clear
and specific questions to gather the information needed to fulfill
the user's request.  Keep in mind that the user may not have all the
necessary details, so you might need to both request additional user
input and call internal tools step by step to reach the goal.
   * answer:  If you choose answer, you must provide an internal tool
call in the content field, with the tool name and its arguments in JSON
format (e.g.  {"name":  tool_name, "arguments":  {"arg_1":  "value_1",
"arg_2":  "value_2"}}).

## Important Notes
   * In each step of interaction, first write your thoughts and
analysis between <think> and </think> to carefully decide your next
step.  Only after providing this reasoning should you call the
interact_with_env tool to interact with the environment.  Always
present your reasoning before making the tool call.
   * The total number of rounds that you can interact with the
environment is limited.  You should smartly {{Placeholder:  Remind the
task's goal}}, so that you can fulfill the user's request in the most
efficient way.
   * Usually you should {{Placeholder:  Hint on the common way to
interact with this gym in order to achieve the task goal}}.
   * Be bold, creative and smart in your interaction with the
environment!  Let's begin!
```

Figure 16: System prompt template for agent before its interaction with the specific gym environment.

