# OpenReview forum: "UserRL: Training Interactive User-Centric Agent via Reinforcement Learning"
_ICLR.cc/2026/Conference — Submitted to ICLR 2026_

### Official Review · Reviewer_zKr9 · 2025-10-29

**Soundness:** 3
**Presentation:** 3
**Contribution:** 3
**Rating:** 6
**Confidence:** 2

**Summary:**

The paper positions itself within the agentic LLM literature. The goal is to develop LLM agents that are more user-centric. Their first contribution is a new gym environment with 8 new tasks that can be used to train and evaluate LLM agents in terms of user assistance. Their pipeline uses LLM-based user simulation and multi-turn RL. Their second contribution is the extension of GRPO to the multi-turn setting by decoupling multi-turn and trajectory level feedback, essentially allowing for different reward distribution among turns. The empirical results with user simulators show improvement over raw LLMs in the proposed gym environments. A proof-of-concept real user study is presented but it is severely limited (5 participants).

**Strengths:**

- UserRL Gym looks like a meaningful contribution for developing/evaluating LLM-based agents. The design seems sensible to me, and the specific environments are interesting. (I am not from the LLM community, so I can’t fully assess novelty.)

- The proposed reward shaping methods look sensible, and the insight provided is useful to understand the design choices.

- The objective—a multi-turn extension of GRPO with decoupled turn-level shaping and trajectory-level scoring—seems like a natural idea and potentially novel.

- Experiments appear strong: the proposed method outperforms raw open-source models and also closed-source models in most tasks. It’s not surprising that Gemini wins in SearchGym, so that result doesn’t bother me.

**Weaknesses:**

- The paper states that the equalized approach uses $\tilde{r}_t = c$ and (as written) this seems to imply that GRPO gives the same reward to every token within a turn. My reading of a recent paper [1] suggests this is a bit more nuanced: in vanilla GRPO, each completion does receive a single outcome-based advantage that’s applied uniformly to all its tokens (in the policy term), but this objective is equivalent to assigning step-level rewards to shared-prefix spans; when prefixes overlap across completions, different steps (and thus tokens) within a turn can receive different effective signals. I’d like the authors to clarify this, preferably in light of [1].

- The analysis of the decoupling (turn-level shaping vs. trajectory-level scoring) could be made more precise, again considering the above nuance from [1].

- Positioning/novelty is hard for me to judge because Related Work is moved entirely to the appendix. The brief intro paragraphs are not enough (for a non-LLM RL reviewer) to place the work clearly.

- Results wording: the statement “Equalized/R2G setting achieves the best performance across 4B and 8B models” should be made precise that this is average performance across all tasks. If we look task-by-task, it loses to other models in 6/8 tasks.

- On user study: as it stands (n=5, all CS PhD students), I would treat this as a proof-of-concept rather than a user study. We cannot really infer anything from this.



[1] GRPO is Secretly a Process Reward Model, 2025,  	arXiv:2509.21154

**Questions:**

1. On the GRPO point: do you really mean GRPO gives the same reward to each token within a turn? How does that reconcile with the GRPO-as-PRM view (shared-prefix/step-level signals) in [1]? If you intend the “uniform per turn” description in the outcome-reward sense, can you please state that explicitly and discuss when shared-prefix structure induces non-uniform token-level signals?

2. For the decoupled objective, in what regimes (task types, group sizes, sampling temperature) does it help beyond what standard GRPO already induces through shared prefixes? Any cases where decoupling hurts?

Please free to respond to any points raised in Weaknesses as well.

[1] GRPO is Secretly a Process Reward Model, 2025,  	arXiv:2509.21154

---

> ### Author Response · Authors · 2025-11-14
>
> > ### **Clarifying the “Equalized” Approach and GRPO Interpretation**
>
> Our *Equalized* setting assigns a *constant turn-level signal* ( \tilde{r}\_t = c ) that is broadcast uniformly to all tokens within a turn before GRPO’s normalization. This follows the **outcome-level uniformity** assumption of vanilla GRPO. We agree with the reviewer that, as shown in the previous work, GRPO’s effective learning signal becomes non-uniform at the token level when shared prefixes exist across completions.
>
> We will further clarify that by *Equalized*, we mean uniform **within a turn** when mapping environment feedback to \tilde{r}\_t = c. This is **outcome-uniform at the turn level**, not a claim that GRPO assigns an identical signal to every token overall. In fact, because GRPO performs group-wise normalization at the trajectory level, and because completions may share prefixes, the effective token-level signals can differ across steps as predicted by the GRPO-as-PRM analysis.
>
> We believe this should address your concern: we did not intend to deny the shared-prefix/step-level effect; our *Equalized* setting was about our shaping interface, not a claim about the *effective* token signals under GRPO.
>
> ***
>
> > ### **Decoupling Turn-Level Shaping and Trajectory Scoring**
>
> Our framework can support the separation of (i) **turn-level shaping** (e.g., Equalized, R2G, EM) from (ii) **trajectory-level scoring** (Sum vs. R2G). This design enables controlled tests of how local versus aggregated feedback affects user-centric agent learning. It does not modify GRPO’s core policy update but restructures how intermediate feedback is assigned. We will further clarify this mathematically and add a short remark linking the implicit step-level signals (from shared prefixes) with our explicit decoupling.
>
> ***
>
> > ### **When Decoupling Helps (and When It May Not)**
>
> **Theoretically,** R2G trajectory scoring should enhance learning in three regimes:
> (1) **Long-horizon or compositional tasks**, where early enabling turns deserve credit despite delayed success;
> (2) **Moderate sampling temperatures and larger groups**, which increase prefix overlap and strengthen GRPO’s implicit step-level (PRM-like) signals; and
> (3) **Sparse-reward settings**, where backward credit propagation stabilizes training.
>
> **Empirically,** we find support for (1) and (3): in *TravelGym*, where tasks require multi-turn composition, decoupled R2G scoring consistently improves performance; and in *FunctionGym*, where rewards appear only at the final step, it mitigates reward sparsity. The second regime remains plausible but not yet confirmed empirically.
>
> We also observe that differentiating per-turn rewards without careful design can sometimes degrade performance or offer little benefit (see original discussion in Section 5). More generally, decoupling tends to be neutral or slightly negative when turn-level rewards are already well-shaped (e.g., *PersuadeGym*) or when prefix overlap is limited, though the latter warrants further empirical study.
>
> ***
>
> > ### **Related Works and Novelty**
>
> We appreciate the reviewer’s concern regarding clarity of originality and comparison. Our main contribution lies in **the design of user-centric gym environments and the flexible testbed enabling controlled RL analyses**.
>
> Please refer to our **first response to Reviewer 1A49** for more details.
>
> ***
>
> > ### **Clarity Refinements**
>
> We will further refine our claim to make the wording precise:
>
> > Equalized/R2G achieves the highest *average performance across all tasks* for both 4B and 8B models.
>
> We will also note that, while it leads in overall mean performance, it does not dominate every task. We will link this to our third response above and current analysis in Section 4.2.
>
> ***
>
> > ### **Broader and Diverse Real-User Evaluation**
>
> We agree that expanding diversity is important. After submission, we scaled the study to **20 participants** (undergraduate and PhD students from CS, Physics, and Economics). Please see the **fifth response (second to last) to Reviewer 7pMg** for details.
>
> ***
>
> We are happy to provide further clarifications and hope our responses adequately address the concerns. If our rebuttal satisfactorily resolves your concerns, we would greatly appreciate your consideration in **reflecting this through an updated score and confidence**. Thank you for your time and insights.

---

> > ### Comment · Reviewer_zKr9 · 2025-11-25
> >
> > Dear authors,
> >
> > Thank you for your detailed response. This addresses some of my concerns. I will increase my confidence, but for now maintain my score.
> >
> > Thanks.

---

> > > ### Author Response · Authors · 2025-11-26
> > >
> > > Thank you very much for your positive assessment of our work! We sincerely appreciate your thoughtful feedback, and we would be delighted to continue the discussion or address any additional questions you may have.

---

### Official Review · Reviewer_KG5K · 2025-10-29

**Soundness:** 3
**Presentation:** 2
**Contribution:** 3
**Rating:** 4
**Confidence:** 4

**Summary:**

This paper proposes UserRL, a framework for training user-centric agents via reinforcement learning (RL) with a focus on dynamic, multi-turn user interactions. A core, compelling idea is its alignment with the realistic scenario where user intentions clarify gradually through interaction—a critical yet underexplored aspect of human-AI collaboration. The work introduces 8 standardized Gym environments covering skills like persuasion, intention understanding, and personalized recommendation, alongside a flexible reward mechanism (Equalized/R2G) and SFT cold-start strategy. Key contributions include: (1) addressing the lack of unified training frameworks for diverse, evolving user behaviors; (2) demonstrating that open-source models trained with UserRL can outperform closed-source counterparts in user interaction tasks; and (3) validating the effectiveness of decoupling turn-level and trajectory-level rewards for multi-turn RL.

**Strengths:**

- **Alignment with Real-World Intent Evolution**: The idea that user intentions *gradually clarify through interaction*—is highly relevant to practical AI assistant use cases.
- **Methodological Clarity**: The design of 8 standardized Gym environments with explicit interaction rules and the detailed breakdown of reward mechanisms (Equalized/R2G, SFT cold-start) provide a reproducible framework for training agents to navigate evolving user intentions.
- **Open-Source Focus**: Demonstrating that open-source models (Qwen3 series) can achieve strong performance in user interaction tasks (surpassing GPT-4o, Gemini-2.5 in some Gyms) highlights accessibility and scalability, especially for researchers studying intention-aware AI.

**Weaknesses:**

- **Ambiguous Capability Boundaries of SFT and RL**: While the paper emphasizes SFT as a “cold start” for RL, it does not clearly delineate which capabilities (e.g., basic interaction norms vs. strategic intention-mining) come from SFT vs. RL. Ablations comparing “SFT-only” vs. “SFT+RL” would clarify how each component contributes to handling *evolving user intentions*.
- **Lack of Zero-Shot Transfer Experiments**: The generalizability of UserRL-trained agents to *unseen* domains (where intention evolution might follow different patterns, e.g., medical consultation) is untested. Zero-shot evaluations on novel Gyms would be critical to assess if the framework truly learns *generalizable strategies* for intention clarification.
- **Impact on General Task Performance**: It is unknown whether the training process degrades performance on foundational benchmarks (e.g., math, code). Evaluating trade-offs between intention-aware interaction skills and general capabilities is necessary to gauge the approach’s practicality.
- **Limited Real-User Scale and Diversity**: Expanding to a larger, more diverse user pool (e.g., non-technical users, different demographics) would strengthen claims about the framework’s ability to handle *naturally evolving intentions* in the wild.

**Questions:**

1. Could you clarify the *exact capabilities* imparted by SFT vs. RL in the context of *evolving user intentions*? For example, please compare performance metrics of “SFT-only” models and “SFT+RL” models in Gyms where intention clarification is central (e.g., IntentionGym, TravelGym) to isolate their respective contributions.
2. What is the performance of UserRL-trained agents in **zero-shot transfer** to entirely new domains (e.g., healthcare or education) where user intentions might evolve differently? Please include these experiments to demonstrate the generalizability of strategies for intention clarification.
3. Does training with UserRL negatively impact performance on *general* benchmarks (e.g., math, code,)? If so, how significant is the trade-off, and does it affect the agent’s ability to support intention-driven tasks (e.g., using math skills to fulfill a user’s budget-related travel needs)?
4. Could you expand real-user testing to a more diverse population (e.g., non-technical users, different age groups) and report on how performance varies when handling *naturally evolving, heterogeneous intentions*?

**Details Of Ethics Concerns:**

See weakness and questions.

---

> ### Author Response · Authors · 2025-11-14
>
> > ### **Clarifying SFT vs. RL Capabilities**
>
> We appreciate the reviewer’s observation on the division of capabilities between SFT and RL. Conceptually, SFT provides a *cold-start foundation* for fundamental interaction norms, such as turn-taking, politeness, and clarification, distilled from GPT-4o trajectories. However, SFT-only models plateau quickly, lacking adaptive mechanisms for evolving user intentions.
>
> Empirically, we add ablations comparing **SFT-only** and **SFT+RL (Equalized/R2G)** variants on **IntentionGym (held-out)**, **TravelGym (held-in)**, and **TurtleGym (held-in)** using **Qwen3-4B**, alongside the *raw* model. All settings share identical SFT datasets and hyperparameters.
>
> | Model | TravelGym | TurtleGym | IntentionGym |
> | -- | -- | -- | -- |
> | Qwen3-4B (Equalized/R2G) | 0.5086    | 0.1844    | 1.8075       |
> | Qwen3-4B (SFT only)      | 0.3921    | 0.1633    | 1.7375       |
> | Qwen3-4B (Raw)           | 0.1405    | 0.0854    | 1.7400       |
>
> SFT markedly improves over the raw model, confirming it **instills stable interaction fundamentals**, yet still trails SFT+RL in our main experiment. SFT also underperforms on held-out tasks, suggesting that memorization without reinforcement may hinder transfer. Qualitatively, we observe that **SFT models can already show emerging user-centric reasoning** (e.g., clarification questions), while **RL tuning generalizes these behaviors** to unseen domains, reflecting genuine *intention-adaptive learning*.
>
> ***
>
> > ### **Zero-Shot Transfer Across Domains**
>
> We note our evaluation already tests **zero-shot transfer** to three unseen gyms (*IntentionGym, TelepathyGym, SearchGym*). As shown in Table 3, Qwen3-8B (Equalized/R2G) achieves competitive or superior results to larger baselines, indicating that learned strategies (e.g., *progressive clarification* and *adaptive questioning*) **generalize effectively** across user goals. Notably, in *IntentionGym*, the trained model exhibits **structured intention clarification**, even in previously unseen scenarios.
>
> While domains like healthcare or education lie outside our current scope, our standardized interface and modular reward design make **extension straightforward**, which we view as future work.
>
> ***
>
> > ### **Impact on General Capabilities**
>
> We thank the reviewer for raising the concern of potential trade-offs. We evaluated **MATH500** and **LiveCodeBench** to measure reasoning and coding skills, comparing trained and raw models.
>
> |                          | MATH500 | LiveCodeBench |
> | -- | -- | -- |
> | Qwen3-4B (Equalized/R2G) | 95.2    | 52.1          |
> | Qwen3-4B (Raw)           | 97.0    | 54.2          |
> | Qwen3-8B (Equalized/R2G) | 96.0    | 54.5          |
> | Qwen3-8B (Raw)           | 97.4    | 57.5          |
>
> Only slight degradation is observed: **UserRL-trained models retain over 95%** of baseline performance. In *TravelGym* and *FunctionGym*, reasoning remains crucial, showing that RL mainly enhances the *integration* of general skills into user-driven contexts. Although minor drops appear in isolated benchmarks, our model’s strength lies in **interactive, intention-evolving tasks**.
>
> ***
>
> > ### **Broader and Diverse Real-User Evaluation**
>
> We agree that expanding diversity is important. After submission, we scaled the study to **20 participants** (undergraduate and PhD students from CS, Physics, and Economics). Please see the **fifth response (second to last) to Reviewer 7pMg** for details.
>
> ***
>
> We are happy to provide further clarifications and hope our responses adequately address the concerns. If our rebuttal satisfactorily resolves your concerns, we would greatly appreciate your consideration in **reflecting this through an updated score and confidence**. Thank you for your time and insights.

---

> > ### Comment · Reviewer_KG5K · 2025-11-26
> > **Thanks for the reply！**
> >
> > Thank you very much for your reply. The additional data effectively addressed my concerns and further demonstrated the necessity and effectiveness of the paper's methodology. After careful consideration, I believe a slight increase in my score is fair.

---

> > > ### Author Response · Authors · 2025-11-26
> > >
> > > Thank you very much for your positive assessment of our work! We sincerely appreciate your thoughtful feedback, and we would be delighted to continue the discussion or address any additional questions you may have.

---

### Official Review · Reviewer_1A49 · 2025-10-29

**Soundness:** 2
**Presentation:** 3
**Contribution:** 2
**Rating:** 6
**Confidence:** 2

**Summary:**

### *Summary*

- The authors propose the UserRL framework, which uses the Interact Tool to encompass multiple gym environments for multi-turn RL with LLM-based user simulation. The authors propose a GRPO-like setup that reconciles trajectory-based advantages for different environments while also including the possibly to have turn-level advantages. The authors include 8 gym environments (3 novel data sources) compatible with UserRL, on which they conduct experiments of various models.

### *Contributions*

- UserRL codebase and framework based on a unified tool interface with 8 gym envs, 3 of which are novel
- An evaluation of various turn-based reward schemes that decouples turn-based reward shaping and trajectory-level rewards
- Evaluation metrics that go beyond simple task success metrics for user-centric RL

(Side note: Appendix B describes the contributions more clearly than the introduction)

**Strengths:**

### *Originality*

- 8 new gym environments, including 3 with newly curated datasets.

### *Quality*

- Interesting results relating to scale and benchmark overfitting

### *Clarity*

- Paper is well-structured, and mostly well-written

### *Significance*

- This interface will be useful to the community if properly implemented and maintained.

**Weaknesses:**

### *Originality*

- Related works in the appendix is always a gamble because it makes it hard for someone who is not deeply familiar with the relevant literature to evaluate the novelty of your work.
- The related work section does not *compare and contrast* to previous work. This makes it hard to evaluate the novelty of the proposed contributions. It would be good to highlight the *differences* with previous works, rather than listing related works.

### *Quality*

- Unclear how easy it is to build on UserRL, either to extend it or to use it in conjunction with e.g. verifiers, or introduce more complex agent-like user simulations
- No validation of the quality of the proposed Gym environments

### *Clarity*

- Figure 1 is disconnected from its caption. For such a complex figure, it’s good to “walk through” the figure and explain what’s going on.
- Weird syntax in some places, e.g. L259 “Importantly, this design reflects user-centric:”. So far I haven’t seen anything that really prevents clear understanding, but these mistakes definitely “break the flow”.
- The position of appendix B is off. This seems like it fits much more at the end of the introduction because it highlights the contributions. Furthermore, the discussion of “novelty” is difficult because it is decoupled from the related works in appendix A.

**Questions:**

The main obstacle for me is understanding how this work positions itself against what is already out there. I am familiar with GRPO, verl, verifiers, etc., but I am not deeply familiar with this literature. There are so many proposed frameworks for RL these days that the distinct value of the framework needs to be extremely clear for me to evaluate its originality, and therefore its significance and quality. Therefore, my main question is:

- What are the closest existing works to UserRL and how is your work different? Why are these differences meaningful?

---

> ### Author Response · Authors · 2025-11-14
>
> > ### **Related Works and Novelty**
>
> We appreciate the reviewer’s concern regarding clarity of originality and comparison. Our main contribution lies in **the design of user-centric gym environments and the flexible testbed enabling controlled RL analyses**.
>
> **(1) Novel Gym Environments.**
> Unlike existing environments such as Tau-Bench, UserBench, and MINT, our eight gyms are designed to **systematically and collectively target a broad spectrum of user-centric abilities**, including intent clarification, persuasion, personalized planning, and tool use. Each gym follows a standardized interaction interface that supports both benchmarking and reinforcement learning training. In contrast to previous benchmark works that primarily offer static evaluation settings and are not directly suitable for training, our environments enable not only evaluation but **interactive, multi-turn learning** that naturally captures **user diversity and dynamic behaviors**.
>
> **(2) Flexible Testbed for Decoupled Reward Design.**
> Prior frameworks such as GRPO and DAPO primarily optimize trajectory-level rewards in single-task or static settings. In contrast, UserRL explicitly decouples **turn-level** and **trajectory-level** reward formulations, enabling empirical analysis of how different reward shaping strategies influence interactive learning. This modular design establishes a **controlled and extensible testbed** for evaluating RL algorithms under dynamic, user-driven feedback. Importantly, our main contribution is **not** to propose a universally optimal reward formulation that achieves SOTA performance across all tasks, but to provide an **empirical platform** that allows researchers to systematically explore and gain insights into user-centric RL. The disentanglement of reward structures is one of the **key features** our framework supports, made possible by the flexible and customizable reward mechanism within our gyms. (Please refer to our **first response to Reviewer 7pMg** for further details.)
>
> **(3) Meaningful Differences.**
> While prior user-centric frameworks such as Proactive Agent focus on specific capabilities or task domains, **UserRL** seeks to **generalize and unify** these efforts within a **reproducible and extensible framework** for both training and evaluation. This unification enables systematic investigation of critical aspects, such as user diversity, interaction dynamics, and credit assignment, across diverse scenarios. In doing so, UserRL serves as a **foundational research platform** rather than a one-off implementation, advancing the study of generalizable user-centric reinforcement learning.
>
> ***
>
> > ### **Extensibility of UserRL**
>
> We agree that extensibility should be clearer. UserRL is designed for **plug-and-play extensions**: researchers can (a) add new gyms through the standardized tool interface, (b) customize reward shaping modules for specific behaviors, and (c) integrate new user simulations or verifiers without modifying the core training loop. For example, building on UserRL can include introducing new reward functions (e.g., verifier-guided feedback) or simulated profiles for user diversity, making it a scalable research platform for user-centric agent training.
>
> ***
>
> > ### **Validation of Gym Quality**
>
> Each gym was designed with explicit task logic and LLM-based user simulations validated through rule-based reward functions. All gym data were created under human supervision and subjected to quality checks to ensure their reliability. Placing these gyms within the broader **UserRL framework**, we assess overall framework quality through **training robustness**:  models trained within our gyms achieve **stable performance gains** over baselines and **transfer effectively to real-user interactions**, demonstrating that the environments produce meaningful learning signals. Furthermore, as detailed in our **fourth response to Reviewer 7pMg**, we conducted an additional experiment by inverting the training and evaluation gyms, which further confirms the **robustness** of our framework. Collectively, these results support the **reliability of UserRL as a research testbed** for user-centric RL.
>
> ***
>
> > ### **Clarity Refinements and Positioning**
>
> We will revise Figure 1 and its captions to provide a clearer, step-by-step walkthrough of the framework, and correct minor syntactic issues throughout the paper to enhance readability without altering technical content. Additionally, we will move the summary of novelty (previously in Appendix B) and the key points emphasized in our first response to the end of the Introduction, making our contributions and their context more transparent.
>
> ***
>
> We are happy to provide further clarifications and hope our responses adequately address the concerns. If our rebuttal satisfactorily resolves your concerns, we would greatly appreciate your consideration in **reflecting this through an updated score and confidence**. Thank you for your time and insights.

---

> > ### Comment · Reviewer_1A49 · 2025-11-20
> > **Reply**
> >
> > Thank you for your reply.
> >
> > The related work section clarifies why your approach is different, and potentially valuable to the community. The accessibility of such a framework that decouples the rewards sounds quite appealing for the community, however it's unclear whether your results substantiate this as making an impactful change (as stated by 7pMg).
> >
> > The validation I had in mind was whether these gyms are of relevant real-world tasks: did you check with any domain experts, or do you consider that these proposed new gyms do not require any such validation, and why?
> >
> > I'll currently keep my score of 6, but up my confidence given your replies.

---

> ### Author Response · Authors · 2025-11-21
> **Follow-up Reply to Reviewer 1A49**
>
> Thank you for the thoughtful follow-up questions. We appreciate the opportunity to further clarify our contributions and positioning.
>
> ---
>
> > ### **On Paper Novelty and the Decoupling Method**
>
> We view decoupling as a mechanism for **flexible reward customization**, enabled by the relatively dense reward signals provided in our gym environments. Within this setting, we demonstrate that trajectory-level reward shaping via *Reward-to-Go* can improve performance on user-centric tasks. This serves as empirical validation that our framework, with fine-grained and composable reward signals, can help foster more effective learning.
>
> While we do not claim our current reward variations to be globally optimal, we emphasize that our design creates **a foundation upon which future work can innovate**. There are many alternative ways to formulate turn-level and trajectory-level rewards, and our contribution lies in enabling such exploration through a high-quality, customizable evaluation and training suite. In this sense, our framework opens the door for further refinement, as also discussed in our earlier responses to Reviewer 7pMg regarding novelty and flexibility.
>
> ---
>
> > ### **On the Relevance of Our Gyms to Real-World Tasks**
>
> The gyms included in our suite target skills such as intention understanding, preference clarification, and other core aspects of user-centric interaction. We view user-centric capability not as an isolated task, but rather as an **essential component** of agents that aim to perform real-world tasks effectively and responsibly. Our emphasis, therefore, is on developing and evaluating abilities that support such real-world performance.
>
> In addition, multiple gyms are adapted from established datasets such as **UserBench** and **Tau-Bench**, both of which are designed to assess user-centric agent abilities. These datasets have been widely used and recognized for their quality, and we believe that integrating them into our gym formulation naturally provides a valid basis for evaluation and benchmarking.
>
> ---
>
> If there are any remaining concerns, we would be glad to further discuss them and provide additional clarification. If our response has addressed your questions, we would also sincerely appreciate your consideration in reflecting this in your evaluations and confidence. Thank you very much for your feedback.

---

### Official Review · Reviewer_7pMg · 2025-10-29

**Soundness:** 2
**Presentation:** 3
**Contribution:** 2
**Rating:** 2
**Confidence:** 4

**Summary:**

The experimental details and gymnasium environments are thorough and well-documented in the main and supplemental material. The gymnasia themselves, while important within the authors' proposed framework, do not appear to be substantially novel contributions (on their own). The authors' argue that their UserRL framework provides flexibility in reward-based training across these gymnasia. Here, the novelty of this framework appears to lie within the turn-level reward shaping. In their Appendix Section B outlining their contributions, this corresponds to point (ii), which seems substantively to be the most important component of this UserRL framework. The authors do include interesting experimental results, particularly the authors' finding that scaling model size does not necessarily yield improvements in their empirical analysis.

**Strengths:**

UserRL introduces a unified and extensible framework for training and evaluating user-centric agents through eight newly designed gym environments and simulated users. Its decoupling of turn-level reward shaping from trajectory-level scoring represents a novel methodological contribution, enabling controlled analysis of multi-turn reinforcement learning (RL) dynamics. The integration of user simulation and standardized tool interfaces provides an innovative approach to studying interactive alignment and reward design. It systematically explores multiple reward-shaping strategies and trajectory-level scoring methods, supporting conclusions with detailed experiments across model sizes and simulators.

**Weaknesses:**

The decoupling of the trajectory and turn based scoring is not sufficiently demonstrated.
Additionally, the authors accurately identify limitations in their Discussion that mitigate the importance of this contribution, without further engineering for heterogenous environments.
While their framework shows promise and the identified gaps/applications are important, this work does not substantially address its motivations. I believe further demonstration of the importance of this novel framework is necessary to justify this as a substantial contribution
Their outlined contributions, while interesting and appreciated, do not sufficiently demonstrate novelty for publication (or at least, are not compellingly justified beyond their statement).

**Questions:**

- As the authors state, this turn-based reward differentiation alone does not appear to be sufficient for a *universal* training strategy. Environment/Gymnasium specific training appears still necessary. This importantly tempers the ability of this work to address some of the authors' proposed motivations, namely that their heterogenous environments address the diversity of user interactions.

- Is there a principled rationale for the selection of reward-shaping? That is, a priori, could a user decide (or even define) gynmasia specific reward-shaping that maintains the authors' proposed "decoupling"? This would appear to mitigate (although not solve) the comment above by providing greater flexibility in training across environments.

- Could the authors' further substantiate their claim that "UserRL generalizes GRPO to multi-turn interactive settings by decoupling turn-level reward shaping from trajectory-level scoring"? Perhaps this follows from the form of their J_{UserRL} equation, but it is currently not obvious from their proposed framework how they achieve this "decoupling". This is a specific but pivotal point, as this claim seems to be the primary proposed contribution of UserRL.

- Can the authors' justify their selection of training and test gyms? Is it posible to simulate independent data for the inverse of their selection (or any other arbitrary configuration of gymnasia)? They may be helpful to demonstrate a sense of robustness of their UserRL framework.

- The sample size for the real-user study (n=5) is underwhelming, and this fact is not apparent in the main text but deep into the Appendix. This should be transparently reported in the main text when discussing the real-user results. Conclusions based on the real-user analysis should be more modest, given this small (and possibly biased, including PhD students who we may presume to be quite familiar end-users) sample. This is not transparently reported.

- Some of the notation is not immediately clear. For example, in the J_{UserRL} equation, $i=1, ..., n$ indexes different trajectories within a single gym (e.g. repeated "experiments"/observations)? It could be useful to explicitly collect some of this notation (i.e. i trajectories, L_t tokens per trajectory time, etc.) rather than stating them throughout the text (space/organization permitting)

- Can the trajectory scorer vary across gymnasia? Or is this fixed across environments?

---

> ### Author Response · Authors · 2025-11-14
>
> > ### **Clarifying the Framework’s Novelty**
>
> We thank the reviewer for the constructive feedback. The novelty of *UserRL* lies not in a single gym or reward variant, but in offering a **comprehensive and flexible testbed** for RL in user-centric, multi-turn settings. The eight gyms form a **diverse substrate** for systematic study of reward design, model behavior, and generalization across interactions.
>
> Our goal is to provide a **reproducible, extensible platform**, not isolated benchmarks or a fixed "best" setup. The disentanglement between turn- and trajectory-level rewards exemplifies this flexibility, showing how different credit structures can be explored within one unified framework. We do not claim this design is optimal, but rather that it demonstrates *UserRL*’s **adaptability** to varied reward definitions, training strategies, and evaluation schemes.
>
> In summary, *UserRL* is a **general infrastructure for flexible RL research**, enabling controlled analysis of design choices and forming a foundation for future extensions by the community.
>
> ***
>
> > ### **Flexibility and Customization Across Gyms**
>
> *UserRL* is **fully customizable**, supporting diverse reward and training configurations:
> (1) Each gym may define dense, sparse, delayed, or omitted turn-level rewards.
> (2) Trajectory-level scoring and aggregation are configurable.
> (3) Scorers may vary across gyms; we fix one per setup for comparability during experiments.
>
> This flexibility supports **environment-specific reward functions** while preserving the decoupling principle. *UserRL* thus serves as a **research sandbox** for controlled exploration of reward structures in user-centric RL.
>
> ***
>
> > ### **Decoupling Mechanism and Relation to GRPO**
>
> In GRPO, trajectory rewards are uniformly propagated to all turns, **entangling local and global signals**. *UserRL* explicitly **separates** these components:
> (1) Each gym defines turn-level shaping;
> (2) The trajectory term evaluates the full interaction as a distinct objective.
>
> This design **extends GRPO to multi-turn settings**, improving interpretability and experimental control. The disentanglement is not a specific shaping strategy but a **framework capability** that supports systematic comparison among strategies without assuming any is universally best.
>
> ***
>
> > ### **Training and Evaluation Design**
>
> Our original train/test split balanced interaction coverage and computational cost. Following the reviewer’s suggestion, we trained *UserRL* (Equalized/R2G, Qwen3-4B) on *TravelGym, IntentionGym, TelepathyGym, FunctionGym,* and *TauGym*, evaluating zero-shot on *SearchGym, TurtleGym,* and *PersuadeGym*:
>
> | **Model** | **TravelGym** | **TurtleGym** | **FunctionGym** | **TauGym** | **PersuadeGym** | **IntentionGym** | **TelepathyGym** | **SearchGym** |
> | -- | -- | -- | -- | -- | -- | -- | -- | -- |
> | Qwen3-4B (original) | 0.5086 | 0.1844 | 0.3333 | 0.2000 | 0.4643 | 1.8075 | 0.6098 | 0.864 |
> | Qwen3-4B (new train) | 0.5054 | 0.1587 | 0.3333 | 0.2121 | 0.4563 | 1.8175 | 0.6341 | 0.864 |
>
> Performance remains **comparable**, confirming robustness to gym composition and suggesting that *UserRL* captures **generalizable RL behaviors** across heterogeneous environments. Future work can explore **inter-gym transfer** to uncover deeper task correlations.
>
> ***
>
> > ### **Human Evaluation and Transparency**
>
> Following submission, we have expanded the study to include **20 diverse participants**: a mix of undergraduate and PhD students from Computer Science, Physics, and Economics backgrounds. We note Background had minimal impact since participants acted purely as users. Updated results are below:
>
> | **Model** | **TurtleGym** | **TelepathyGym** |
> | -- | -- | -- |
> | Qwen3-4B (GPT-4o User) | 0.1844 | 0.6098 |
> | Qwen3-4B (Real User) | 0.2631 | 0.7073 |
> | Qwen3-8B (GPT-4o User) | 0.1854 | 0.5610 |
> | Qwen3-8B (Real User) | 0.2856 | 0.7317 |
>
> Despite score variance, **the trend is consistent**: real-user evaluations outperform simulated ones, supporting our claim that **human interactions yield more cooperative dynamics**. These results will be added for transparency and stronger empirical grounding.
>
> ***
>
> > ### **Clarifying Notation and Presentation**
>
> We thank the reviewer for noting this. In \(J_{\text{UserRL}}\), index \(i\) denotes **distinct trajectories within a single gym**, representing repeated trials under identical input conditions. We will clarify this in the revision.
>
> ***
>
> We are happy to provide further clarifications and hope our responses adequately address the concerns. If our rebuttal satisfactorily resolves your concerns, we would greatly appreciate your consideration in **reflecting this through an updated score and confidence**. Thank you for your time and insights.

---

> > ### Author Response · Authors · 2025-11-27
> >
> > Dear Reviewer 7pMg,
> >
> > Thank you very much for your thoughtful comments. Following your suggestions, we have **added further results, made our explanations, and responded point by point to your concerns**. We hope these revisions clarify our contribution and address the issues you raised.
> >
> > As other reviewers have expressed **positive assessments**, we would be truly grateful to know whether our rebuttal also resolve your concerns as **the discussion period is coming to an end**. Thank you again for your time and constructive feedback, and wish you a happy Thanksgiving.
> >
> > With gratitude,
> > Authors of Paper 8034

---

### Author Response · Authors · 2025-11-25

Dear Reviewers,

Thank you for taking the time to evaluate our submission. We have now posted our responses to the comments and would welcome any further feedback. If there are points that would benefit from additional clarification, we would be very glad to provide more detail.

If our answers sufficiently address your concerns, we also kindly ask that you consider reflecting this in your evaluation. We sincerely appreciate your efforts and your support throughout the review process.

With gratitude,
Authors of Paper 8034

---

### Author Response · Authors · 2025-12-01
**Summary of Rebuttal Outcomes and Final Assessment**

Dear Area Chair,

Thank you for your time and effort in reviewing our submission. Following the recommendation of the ICLR 2026 PC, we would like to provide a brief factual summary of the rebuttal for this paper. Our goal is to accurately reflect the reviewer’s engagement during the process, which we greatly appreciate.

Before the scores were reverted, **three reviewers assigned an overall positive score of 6** after reading our rebuttal, with **two of them also increased their confidence** in this positive assessment. The only reviewer who maintained a negative score did not respond to our rebuttal.

---

**Reviewer zKr9**

The reviewer initially gave a positive score of 6 and raised several technical concerns regarding our interpretation of the “Equalized” setting, its alignment with the GRPO-as-PRM view, and the conditions under which decoupling turn-level shaping from trajectory-level scoring is beneficial. In our rebuttal, we addressed these points by:
- clarifying that Equalized applies a constant turn-level signal while remaining consistent with GRPO’s shared-prefix, non-uniform token-level behavior;
- explaining how our framework cleanly separates turn-level shaping from trajectory-level scoring without altering GRPO’s core update;
- identifying theoretical and empirical regimes where decoupling is helpful (e.g., TravelGym, FunctionGym) and where its effect is neutral or slightly negative;

After reviewing these clarifications, the reviewer explicitly **increased their confidence to 3** while maintaining their positive score of 6.

---

**Reviewer KG5K**

The reviewer gave an initial score of 4 and highlighted several key concerns regarding the division of capabilities between SFT and RL, generalization to unseen domains, potential degradation of general capabilities, and the limited scale and diversity of the real-user study. In our rebuttal, we addressed these points by:
- providing new ablations comparing SFT-only, SFT+RL (Equalized/R2G), and raw models on TravelGym, TurtleGym, and IntentionGym, clarifying which capabilities each component imparts;
- explaining that our evaluation already includes zero-shot transfer to three unseen gyms and discussing how the framework can extend to new domains;
- presenting MATH500 and LiveCodeBench results showing only slight degradation relative to raw models, with core reasoning capabilities preserved;
- noting that our real-user study was expanded post-submission to 20 participants with broader academic backgrounds.

After reviewing these additions, the reviewer wrote that the new data “effectively addressed my concerns” and **raised the overall evaluation to 6 with a confidence of 4**, representing a positive assessment.

---

**Reviewer 1A49**

The reviewer initially scored the paper a 6, and they raised concerns about the clarity of our novelty, the positioning of UserRL relative to prior work, the extensibility and validation of our gym environments, and several presentation issues. In our rebuttal, we addressed these by:
- clearly distinguishing UserRL from existing frameworks through its unified interface, novel gym environments, and modular reward decoupling design;
- explaining the extensibility of the framework and how new gyms, reward modules, and user simulators can be integrated;
- detailing the design and quality validation of the gym environments, including their grounding in established datasets and supervised creation process;

After reviewing these clarifications, the reviewer stated that the related-work positioning and explanations were helpful and **increased their confidence to 3** while maintaining their original positive score of 6.

---

**Reviewer 7pMg**

The reviewer initially gave a negative score of 2 and raised concerns about the novelty of the framework, the justification and demonstration of the decoupling mechanism, the choice of training/testing gyms, and the limitations of the real-user study. In our rebuttal, we addressed these concerns by:
- clarifying that UserRL’s contribution lies in providing a unified, extensible testbed for multi-turn user-centric RL;
- explaining the decoupling mechanism in detail and demonstrating its flexibility through configurable turn-level shaping and trajectory-level scoring;
- presenting new experiments with an inverted train/test gym setup showing robustness across compositions;
- expanding the real-user study to 20 participants and reporting updated results.

Despite these targeted and substantial clarifications, **the reviewer did not respond further** during the discussion period.

---

We hope this summary reflects our appreciation for the reviewers’ engagement and the improvements enabled by the discussion. With **three reviewers confirming positive scores before the score reversion and two increasing their confidence**, we hope you will find it suitable for acceptance.

With gratitude,
Authors of Paper 8034

---

### Meta-Review · Area_Chair_oCTo · 2026-01-02

**Summary:**

UserRL presents a framework for training user-centric LLM agents via RL, contributing eight gym environments and a configurable reward structure that decouples turn-level shaping from trajectory-level scoring. Reviewers raised concerns about novelty, insufficient demonstration of the decoupling mechanism's benefits, limited real-user validation, and unclear positioning. Two reviewers converged on marginally positive scores (6), one suggested increasing their original score (4), while one remained strongly negative (2) without engaging in discussion.

This submission is fundamentally a systems/benchmark paper rather than a core ML contribution. The algorithmic component (GRPO + reward shaping variants) is incremental, and the primary value lies in the testbed itself. The validation that this testbed produces genuinely user-centric (not simulator-centric) agents is insufficient. For a systems paper, demonstrating real-world utility is crucial, and this gap is concerning.

**Reviewer Concerns:**

- Positioning relative to prior work clarified (1A49)
- GRPO interpretation and Equalized setting semantics clarified (zKr9)
- Real-user study expanded from 5 to 20 participants


- Algorithmic novelty: The paper applies existing GRPO to new environments; the "decoupling" is a design heuristic rather than a principled algorithmic contribution
- Simulator bias: Insufficient evidence that learned behaviors transfer to real humans rather than exploiting LLM simulator patterns; the expanded user study (20 academic participants) remains limited in scale and diversity
- Turn-level reward limitations: As authors acknowledge, current formulations cannot distinguish insightful from irrelevant agent actions—a significant gap for a paper emphasizing reward design
- 7pMg's concerns about fundamental novelty went unaddressed due to reviewer non-engagement

**Reviewer Scores:**

- Reviewer zKr9: Would maintain 6 with increased confidence (satisfied with clarifications)
- Reviewer KG5K: Confirmed increase (stated concerns were "effectively addressed")
- Reviewer 1A49: Would maintain 6 with increased confidence
- Reviewer 7pMg: Unknown (did not respond). Core novelty concerns may have persisted at 2-4.

---

### Decision · Program_Chairs · 2026-01-26

Reject